# Towards Flexible and Controllable Unknown Rejection

## Abstract

Reliable prediction is an essential requirement for deep neural models that are deployed in open environments, where both covariate and semantic out-of-distribution (OOD) data arise naturally. Recent studies have formulated and pursued two problems named OOD generalization and detection independently, where the former aims to correctly recognize covariate shifts while the latter focuses on rejecting semantic shifts. However, existing methods are misaligned with real-world applications in two aspects. First, in practice, to make safe decisions, a reliable model should accept correctly recognized inputs while rejecting both those misclassified covariate-shifted and semantic-shifted examples. Second, considering the potential existing trade-off between rejecting different failure cases, more convenient, controllable, and flexible unknown rejection approaches are needed. To meet the above requirements, we propose a novel and elegantly simple unknown rejection framework to unify and facilitate classification with rejection under both covariate and semantic shifts. Our key insight is that by separating and consolidating failure-specific reliability knowledge with low-rank adapters and then integrating them, we can enhance the unknown rejection ability effectively and flexibly. Extensive experiments demonstrate the superiority of our framework.

## 1 Introduction

Deep neural models have achieved remarkable performance in closed-world scenarios, assuming that train and test sets come from the same distribution. However, in practice, out-of-distribution (OOD) data naturally arises during the deployment (Nguyen et al., 2015), which mainly includes two types named *covariate shifts* and *semantic shifts* (Bai et al., 2023). Specifically, as depicted in Fig. 1, a model trained on in-distribution (ID) data may encounter covariate shifts such as conditions with snowy night (Sakaridis et al., 2021) or corrupted inputs resulting from camera noise and sensor degradation (Hendrycks & Dietterich, 2018). Unfortunately, the model often suffers significant performance deterioration when deployed in those scenarios. To ensure safety, it is expected to reject wrong predictions instead of accepting them blindly. Alternatively, unknown categories with semantic shifts may also emerge (Hendrycks & Gimpel, 2016; Hendrycks et al., 2018). In this case, the model must reject to make incorrect decisions by detecting semantic-shifted examples.

In recent years, both covariate and semantic shifts have received extensive attention, and have been formulated as OOD generalization (Hendrycks et al., 2021; Yi et al., 2021; Liu et al., 2021; Schneider et al., 2020) and detection (Hendrycks et al., 2018; Zheng et al., 2024; Liu et al., 2020; Basart et al., 2022) problems, respectively. Concretely, the former focuses on recognizing inputs with covariate shifts while the latter focuses on rejecting inputs with semantic shifts. Instead of pursuing those two problems independently, Bai et al. (2023) handles OOD generalization and detection simultaneously by leveraging unlabeled wild data consisting of both covariate and semantic shifts during training. However, the aforementioned efforts still have primary limitations. First, for OOD generalization, there is no rejection option involved, and accepting misclassified covariate-shifted inputs could lead to catastrophic issues. Second, for OOD detection, the performance of prevalent methods drops a lot when inputs of known classes suffer from covariate shifts, and rejecting semantic-shifted samples while accepting all covariate-shifted samples may also lead to serious safety issues.

In addition, the trade-off between the rejection of different failure sources further complicates the problem. Recent studies (Jaeger et al., 2022; Kim et al., 2023b; Narasimhan et al., 2024)

Figure 1: (a) Unknown rejection rejects both the (✗) misclassified covariate-shifted and all semantic-shifted OOD samples, and accepts the (✓) correct prediction. (b) Illustration of three types of common failure cases in the natural open environment.

have observed that prevalent OOD detection methods proposed in the literature often sacrifice the performance when detecting incorrect predictions of ID samples. There are a few studies (Zhu et al., 2024a; Cen et al., 2023; Zhu et al., 2023a; Li et al., 2024) focused on developing reliable models that can reject both misclassified ID and semantic-shifted OOD data. Nevertheless, they typically overlook covariate-shifted samples, and it is hard to distinguish correct covariate-shifted samples from semantic-shifted ones. Besides, they typically train a deep model from scratch or fully fine-tune one, which is computationally heavy and inefficient. In practical scenarios, ***different failure sources are not always predefined and can emerge continually***. For instance, an autonomous driving system performs classification with misclassification rejection on ID data under a normal environment (e.g., clean inputs on a sunny day), and switches to more challenging unknown rejection under covariate shifts when facing sensory degeneration or bad weather (Fig. 1 (b, Middle)). Moreover, when a car drives into the countryside, it may encounter unexpected novel objects such as sheep and deer (Fig. 1 (b, Right)), where the model should perform OOD detection and make a warning. In more common situations, a model is expected to have good rejection ability on various failure cases in the wild without reliability disparity. From a multi-objective optimization perspective, we could simultaneously optimize the model with existing methods dealing with covariate and semantic shifts. However, it is often hard or impossible (Kendall et al., 2018; Boyd & Vandenberghe, 2004) to find a single optimal solution that can optimize the performance on different failure sources simultaneously. Moreover, a single prefixed, static solution lacks the flexibility to explore and calibrate the trade-off among different requirements. Therefore, there is a demand for developing ***flexible*** and ***controllable*** unknown rejection methods.

The goal of this paper is to show that the above-mentioned limitations and requirements can be considerably addressed. For one thing, we aim to predict and accept correctly classified covariate-shifted examples while rejecting those misclassified ones and all unknown samples with semantic shifts. As illustrated in Fig. 1 (a), unlike the OOD detection problem that defines "positive" and "negative" with regard to the label space, unknown rejection directly specifies the distinction by the correctness of model's predictions, which is more reasonable and aligned with the requirement in practical applications. For another, considering the trade-off between rejecting different failure sources, we aim to develop a more flexible method that enables us to easily separate, consolidate, and incorporate different reliable knowledge regarding surrounding environments.

**Contributions.** (1) We study the unknown rejection problem under both covariate and semantic shifts, and call for flexible and controllable methods for reliability enhancement. (2) We propose a reliability arithmetic framework with low-rank adapters to compress and consolidate reliability knowledge effectively and flexibly. To the best of our knowledge, this work is the first to separate and compress reliability knowledge via low-rank adapters. Further, a random projection strategy is proposed for rank adaptation to enhance the tuning efficiency. (3) Comprehensive experiments demonstrating the strong performance of our method, as well as the flexibility of reliability edition.

## 2 PROBLEM FORMULATION

**Training on in-distribution data.** We focus on the multi-class classification setting. Let $\mathcal{X} \subset \mathbb{R}^d$ be an input space, $\mathcal{Y} = [K] = \{1, ..., K\}$ denotes the label space and $\mathcal{P}_{\text{in}}$ be the underlying in-distribution (ID) over $\mathcal{X} \times \mathcal{Y}$. Given a labeled training set $\mathcal{D}_{\text{in}}^{\text{train}} = \{(\mathbf{x}_i, y_i)\}_{i=1}^N$ comprising $N$

samples drawn *i.i.d.* from the joint data distribution $\mathcal{P}_{\text{in}}$, multi-class classification aims to learn a classifier $h : \mathcal{X} \rightarrow \mathcal{Y}$ with low misclassification error. Typically, we learn a function $f : \mathcal{X} \rightarrow \mathbb{R}^K$ that yields the posterior distributions of a given input by minimizing an empirical surrogate risk, e.g., cross-entropy (CE) loss, on $\mathcal{D}_{\text{in}}^{\text{train}}$, and then $h(\mathbf{x}) = \arg\max_{y \in [K]} f_y(\mathbf{x})$.

**Inference in open environments with wild data.** Trained on the ID data, a classifier $f$ deployed in open environments can encounter various out-of-distribution (OOD) shifts, as shown in Figure 1(a). Typically, the OOD data can be grouped into covariate and semantic shifts (Yang et al., 2021):

- Covariate OOD $\mathcal{P}_{\text{out}}^{\text{covariate}}$ has the same label space $\mathcal{Y}$ as the training data, but the input space $\mathcal{X}^{\text{covariate}} \subset \mathbb{R}^d$ undergoes shifting and therefore is different from $\mathcal{X}$.
- Semantic OOD $\mathcal{P}_{\text{out}}^{\text{semantic}}$ represents new-class shifted samples that do not belong to any known classes, i.e., $y \notin \mathcal{Y}$. We further assume that the input space $\mathcal{X}^{\text{semantic}}$ and $\mathcal{X}$ are also in different subsets of $\mathbb{R}^d$, which makes OOD detection possible.

For inference with covariate shifts, existing literature formulates the OOD generalization problem (Hendrycks et al., 2021; Yi et al., 2021; Liu et al., 2021; Schneider et al., 2020) which aims to improve the classification accuracy of covariate-shifted samples. For inference with semantic shifts, prior studies formulate the OOD detection problem (Hendrycks et al., 2018; Zheng et al., 2024; Liu et al., 2020; Basart et al., 2022) which focuses on separating ID and semantic OOD.

**Formulation of unknown rejection in the wild.** In practice, one is likely to encounter both types of samples during classifier deployment. To this end, unknown rejection allows for abstention on both misclassified covariate-shifted and semantic-shifted data, while only accepting correctly classified inputs from known classes ($y = h(\mathbf{x})$ and $y \in \mathcal{Y}$). Formally, considering all possible distributions that a model may encounter in practice, we suppose the test distribution $\mathcal{P}^{\text{test}}$ is a mixture of data from in-distribution, covariate-shifted and semantic-shifted distributions:

$$\mathcal{P}^{\text{test}} = (1 - \pi_{\text{c}} - \pi_{\text{s}})\mathcal{P}_{\text{in}} + \pi_{\text{c}}\mathcal{P}_{\text{out}}^{\text{covariate}} + \pi_{\text{s}}\mathcal{P}_{\text{out}}^{\text{semantic}}, \tag{1}$$

where $\pi_{\text{c}}, \pi_{\text{s}}, \pi_{\text{c}} + \pi_{\text{s}} \in [0, 1]$. The goal of unknown rejection is to learn the classifier $h$ and design a rejector $r : \mathbb{R}^d \rightarrow \{0, 1\}$, where an ideal rejector can ensure to make safe decisions by separating correctly classified samples from misclassified ones or semantic OOD data as follows:

$$r(\mathbf{x}) = \begin{cases} 1 & \text{if } \mathbf{x} \in \mathcal{P}_{\text{in}}(y \neq h(\mathbf{x})) \cup \mathcal{P}_{\text{out}}^{\text{covariate}}(y \neq h(\mathbf{x})) \cup \mathcal{P}_{\text{out}}^{\text{semantic}} \\ 0 & \text{if } \mathbf{x} \in \mathcal{P}_{\text{in}}(y = h(\mathbf{x})) \cup \mathcal{P}_{\text{out}}^{\text{covariate}}(y = h(\mathbf{x})) \end{cases}. \tag{2}$$

Here we emphasize the distinction between those three problems introduced above. OOD generalization only focuses on classification accuracy and has ***no rejection option***; OOD detection ***only rejects*** semantic-shifted samples from unknown classes ($y \notin \mathcal{Y}$), and blindly accepts misclassified samples from known classes ($y \neq h(\mathbf{x})$ and $y \in \mathcal{Y}$). Besides, misclassification detection (MisD) focuses on known classes and rejects misclassified ones. Unknown rejection provides a unified classification with rejection framework that satisfies the practical requirements.

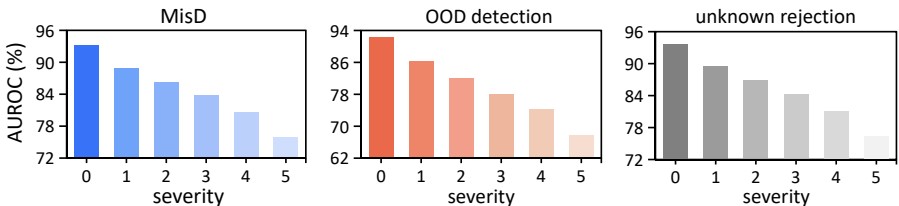

Figure 2: Covariate shifts remarkably complicate the problem of unknown rejection.

**Unknown rejection in the wild is quite challenging.** Prior works (Hendrycks & Gimpel, 2016; Hendrycks et al., 2018; Liu et al., 2020; Cen et al., 2023) often study the rejection ability of a model without considering covariate shifts that will be anticipated at inference time. Actually, unknown rejection under covariate shifts is quite difficult. As shown in Fig. 2 (ResNet-18 (He et al., 2016) trained on CIFAR-10): (1) Within known classes, covariate shifts make it much harder to separate misclassified examples from correct ones. When increasing the corruption severity, the performance of MisD continually drops. (2) Considering OOD detection performance, the model struggles to

distinguish between known and unseen classes when the samples of known classes undergo covariate shifts. (3) From (1)-(2), we know that $\mathcal{P}_{\text{out}}^{\text{covariate}}(y \neq h(\mathbf{x}))$ and $\mathcal{P}_{\text{out}}^{\text{covariate}}(y = h(\mathbf{x}))$ are hard to be separated, and the confidence distributions of $\mathcal{P}_{\text{out}}^{\text{covariate}}$ and $\mathcal{P}_{\text{out}}^{\text{semantic}}$ are also mixed. Therefore, it is quite challenging to achieve the goal of unknown rejection in Eq. (2).

## 3 THE PROPOSED FRAMEWORK: TRUSTLoRA

### 3.1 MOTIVATION

**Limitation of failure-specific full training.** It is acknowledged that rejecting incorrect predictions is essential for reliable learning. However, the failure sources are rich in uncontrolled environments, including incorrect predictions of ID or corrupt-shifted samples, and also inputs from unknown new categories. Current methodologies predominantly focus solely on rejecting one specific failure case, e.g., OOD detection only rejects data with semantic shifts while accepting all other samples. This paradigm, however, has evident limitations: **(1)** *Unrecoverable.* Enhancing the ability of rejection on one specific failure may lead to unrecoverable damage on other aspects of the model, since it has been empirically revealed that trade-off existed when rejecting different failure sources (Jaeger et al., 2022; Kim et al., 2023b). This is undesirable in practice: an autonomous car can not return to its "standard" mode for normal environment after full tuning in OOD environment. **(2)** *Inflexible.* Full training with failure-specific optimization objectives often leads to a static solution. Considering the complexity of open environments, it is beneficial to have convenient ways that can flexibly adjust the trade-off at inference time without full retraining. **(3)** *Inefficient.* When facing new failure cases, full training a model is computationally intensive and time-consuming. In practice, to avoid catastrophic consequences, we expect the model to handle novel failure sources with minimal overhead in latency.

**Reliability knowledge separation and integration.** With the above limitations in mind, we propose to develop unknown rejection framework with separable and combinable reliability knowledge, which is different remarkably from the prior efforts. As demonstrated by Gueta et al. (2023), knowledge can be represented by a region in weight space. Our high-level idea is to compress reliability knowledge regarding different failure cases and then selectively integrate them based on real-world requirements. To this end, two important questions arise: how to get failure-specific knowledge and how to compress it. **(1)** *Acquire reliability.* Many methods have been developed in recent years for reliable prediction, and they often excel at one specific failure case. Those methods form a rich and diverse toolbox, which can be interpreted as encapsulating the specific reliability knowledge naturally. **(2)** *Compress reliability.* Common strategies to compress knowledge such as pruning (Tanaka et al., 2020) and knowledge distillation (Hinton et al., 2015) often suffer from the heavy computation issue, which conflicts with the efficient principle. Therefore, we hope to compress knowledge to a small set of parameters, enabling cheap computation and lightweight integration.

Based on the above discussion, we propose a novel **TrustLoRA** framework to acquire and integrate trustworthy knowledge, which is illustrated in Fig. 3 and detailed below.

### 3.2 RELIABILITY KNOWLEDGE SEPARATION WITH LOW-RANK ADAPTATION

**LoRA-adapted reliability acquiring.** To acquire and compress specific reliable knowledge related to covariate shits, we propose to fine-tune the model in specific low-rank subspace. Concretely, we leverage parameter efficient tuning technique with an auxiliary low-rank adapter (LoRA) (Hu et al., 2021). As illustrated in Fig. 3, LoRA composes of two rank decomposition matrices $\mathbf{B} \in \mathbb{R}^{u \times r}$ and $\mathbf{A} \in \mathbb{R}^{r \times v}$ where $r \in \mathbb{N}$ is the rank and $r \ll \min(u, v)$. $v$ and $u$ are the dimensionality of the input $\hat{\mathbf{x}} \in \mathbb{R}^v$ for current layer and hidden features, respectively. Therefore, $\mathbf{BA} \in \mathbb{R}^{u \times v}$ has the same size as the parameters, i.e., $\mathbf{W} \in \mathbb{R}^{u \times v}$, of the corresponding fully-connected layer in the feature extractor. The modified forward pass with LoRA becomes:

$$\mathbf{z} = (\mathbf{W} + \mathbf{BA})\hat{\mathbf{x}} = \mathbf{W}\hat{\mathbf{x}} + \mathbf{BA}\hat{\mathbf{x}}, \tag{3}$$

where $\mathbf{z} \in \mathbb{R}^u$ is the output, which will be the input of the next layer after passing non-linear activation. During the training stage, the original parameters $\mathbf{W}$ remain frozen, while only $\mathbf{A}$ and $\mathbf{B}$ are trainable, which is low-cost and parameter efficient.

To acquire and separate reliable knowledge in dynamic open environments, we propose to optimize the failure-specific objectives via the LoRA branch as follows. In this work, we follow most of

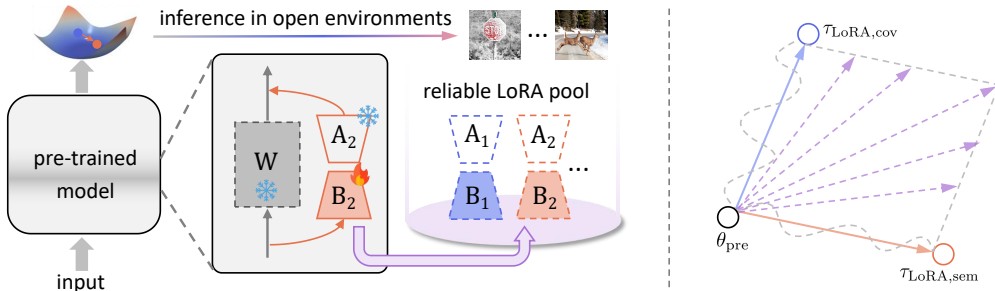

Figure 3: Illustration of the proposed reliability arithmetic framework. (Left) We freeze the pre-trained backbone and add a LoRA module to acquire failure-specific knowledge. (Right) The LoRAs are stored in the memory, and will be merged via arithmetic for unified unknown rejection in the wild.

existing studies that assume the real OOD data is unavailable. For covariate shifts, we leverage AugMix (Hendrycks et al., 2019b), which is a simple augmentation method with the following learning objective:

$$\mathcal{L}_{\text{LoRA,cov}} = \mathcal{L}_{\text{CE}}(f(\mathbf{x}), y) + \lambda \text{JS}\left(f(\mathbf{x}); f(\mathbf{x}_{\text{augmix1}}); f(\mathbf{x}_{\text{augmix2}})\right). \tag{4}$$

Denote $\overline{f} = (f(\mathbf{x}) + f(\mathbf{x}_{\text{augmix1}}) + f(\mathbf{x}_{\text{augmix2}}))/3$ the averaged posterior distributions of $\mathbf{x}$ and its augmented variants, and then the JS loss is: $\text{JS}\left(f(\mathbf{x}); f(\mathbf{x}_{\text{augmix1}}); f(\mathbf{x}_{\text{augmix2}})\right) = \frac{\lambda}{3}\left(\mathcal{L}_{\text{KL}}(f(\mathbf{x}), \overline{f}) + \mathcal{L}_{\text{KL}}(f(\mathbf{x}_{\text{augmix1}}), \overline{f}) + \mathcal{L}_{\text{KL}}(f(\mathbf{x}_{\text{augmix2}}), \overline{f})\right)$. For semantic shifts, we use OE (Hendrycks et al., 2018), which helps the model acquire the knowledge of unknown classes by introducing auxiliary outliers $\mathcal{D}_{\text{aux}}$. Specifically, we minimize the following objective:

$$\mathcal{L}_{\text{LoRA,sem}} = \mathcal{L}_{\text{CE}}(f(\mathbf{x}), y) + \lambda \cdot \mathcal{L}_{\text{KL}}\left(f(\mathbf{x}_{\text{aux}}), \mathcal{U}([K])\right), \tag{5}$$

where $\lambda > 0$ is a scalar, $\mathcal{U}([K])$ represents the uniform distribution over the training label space $\mathcal{Y} = [K]$. $\{\mathbf{A}, \mathbf{B}\}$ denotes all trainable parameters. Since the pre-trained backbone is frozen, the newly added LoRA captures the residual knowledge regarding the specific learning objectives.

**Remark.** We would like to clarify that we do not propose novel failure-specific learning objectives in this paper. Instead, we focus on designing a unified framework to integrate different sources of reliability knowledge in a flexible and parameter-efficient manner.

**LoRA with random projection.** For the initialization of LoRA, the common way is to initialize $\mathbf{B}$ with an all-zero matrix, while initialize $\mathbf{A}$ with a normal distribution. Specifically, each element in $\mathbf{A}$ is independently sampled from a standard Gaussian distribution. In other words, LoRA first projects the input $\hat{\mathbf{x}}$ into a low-rank space via random projection, and then decodes it to the original space. For random projection, the Johnson-Lindenstrauss (Dasgupta & Gupta, 2003) states that the pairwise relation between any two data points can be preserved in an appropriate lower-rank space. Therefore, we further fix the parameters of $\mathbf{A}$ once initialized and only optimize $\mathbf{B}$ in a LoRA module during the training stage, which is much more efficient than learning the original LoRA. Besides, we can store the random seed that generates the random projection of $\mathbf{A}$, requiring much less memory than storing the full matrix, as shown in Fig. 3. We empirically verify that LoRA only introduces a quite small amount of extra trainable parameters that are less than 1% of the original parameters.

### 3.3 RELIABILITY KNOWLEDGE CONSOLIDATION WITH LORA ARITHMETIC

Let $\theta_{\text{pre}} \in \mathbb{R}^M$ be the parameters of a given pre-trained model, where $M$ is the number of parameters. In order to deal with unknown rejection in the wild, we freeze $\theta_{\text{pre}}$ and learn an additional LoRA module with a loss function related to a specific emerged failure at phase $t$ (e.g., OE loss for semantic shifts). Let $\theta_{\text{LoRA},t-1} \in \mathbb{R}^m$ be the weights of the LoRA before fine-tuning, $\theta_{\text{LoRA},t} \in \mathbb{R}^m$ be the corresponding weights after fine-tuning and $m \ll M$. The LoRA vector $\tau_{\text{LoRA},t}$ is given by the element-wise difference between $\{\theta_{\text{pre}}, \theta_{\text{LoRA},t}\}$ and $\{\theta_{\text{pre}}, \theta_{\text{LoRA},t-1}\}$ as follow:

$$\tau_{\text{LoRA},t} = \{\theta_{\text{pre}}, \theta_{\text{LoRA},t}\} - \{\theta_{\text{pre}}, \theta_{\text{LoRA},t-1}\} = \theta_{\text{LoRA},t} - \theta_{\text{LoRA},t-1}. \tag{6}$$

The intuition behind LoRA vector is to encapsulate crucial directions in which the model's parameters move when learning with a loss function (Ilharco et al., 2022) dealing with a specific failure source.

As illustrated in Fig. 3 (Right), after fine-tuning each LoRA module with its respective learning objective, we can perform reliability enhancement or reduction easily and flexibly via element-wise addition or negation with a scaling term $\alpha \in [0, 1]$ as follows:

- **LoRA addition**. The sum of the LoRA vectors $\tau = \sum_t \alpha_t \tau_{\text{LoRA},t}$ is added to a pre-trained model $\theta_{\text{pre}}$ to produce a model that performs unknown rejection on different failure sources. In our cases, we focus on model reliability under both covariate and semantic shifts, and we can get a model $\{\theta_{\text{pre}}, \tau\}$ with unified unknown rejection ability by merging the two LoRA vectors trained using AugMix and OE easily, in which

$$\tau = (1 - \alpha) \cdot \tau_{\text{LoRA,cov}} + \alpha \cdot \tau_{\text{LoRA,sem}}. \quad (7)$$

- **LoRA negation**. We can reduce the ability of rejecting specific failure while retaining performance in other cases by subtracting the LoRA vector from the given LoRA-augmented model. For example, we can get a model $\{\theta_{\text{pre}}, \tau\}$, whose OOD detection ability is weaken with $\tau = -\alpha \cdot \tau_{\text{LoRA,sem}}$.

The LoRA arithmetic is simple and effective to address the challenging unified unknown rejection. Specifically, in our case, we get LoRA vectors regarding covariate and semantic shits via learning objectives presented in Eq. (4) and Eq. (5), respectively. Then we perform LoRA addition to consolidate those two aspects of reliability. The proposed LoRA arithmetic has the following advantages: **(1)** *Flexible.* the scaling term $\alpha$ provides the possibility and flexibility to control the strength of reliability edition, easily adjusting the trade-off without full retraining. **(2)** *Efficient.* When facing new failure cases, we only fine-tune the LoRA, which is lightweight and computationally efficient with minimal latency compared with full training. **(3)** *Recoverable.* We can easily recover the model to the default setting without losing the original knowledge by removing the LoRA module.

**Theoretical analysis.** The investigated problem involves dealing with multiple failure cases, which can be formulated as a multi-objective learning problem. Recently, it has been proved that a linear combination of multiple base models can lead to a pareto-optimal solution with diverse preferences (Dimitriadis et al., 2023). In our work, we build on a pre-trained base model with parameters and introduce LoRA vectors to capture and compress the failure-specific reliable knowledge. Based on the Theorem in Dimitriadis et al. (2023), we can state the approximation power of the proposed LoRA arithmetic as the following Proposition, which states that TrustLoRA can flexibly find a model with a controllable solution for any scaling term $\alpha \in [0, 1]$. The proof can be found in the Appendix.

**Proposition 3.1.** *Given a compact $\mathcal{X} \subseteq \mathbb{R}^D$ and a family of continuous mappings $f_n : \mathcal{X} \to \mathbb{R}^{D'}$, $n = 1, \ldots, N$, there exists a ReLU multi-layer perceptron $f$ with base parameters $\theta_{pre}$ and two low-rank vectors $\tau_{\text{LoRA,cov}}$ and $\tau_{\text{LoRA,sem}}$, such that for any $\epsilon > 0$ and all $n$, there exists an $\alpha \in [0, 1]$ satisfying $\|f_n(x) - f(x; \theta_{pre} + (1 - \alpha) \cdot \tau_{\text{LoRA,cov}} + \alpha \cdot \tau_{\text{LoRA,sem}})\| \le \epsilon$ for all $x \in \mathcal{X}$.*

## 4 EXPERIMENTS

**Datasets and implementation.** Following the common setup in literature, we assume that the real distribution of OOD data remains unknown during training. For covariate-shifted data, we use CIFAR-10/100-C (Hendrycks & Dietterich, 2018) consists of 15 diverse corruption types; for semantic-shifted data, we use natural image datasets including SVHN (Netzer et al., 2011), Textures (Cimpoi et al., 2014), Places (Zhou et al., 2018), LSUN-Crop (Xu et al., 2015), LSUN-Resize (Yu et al., 2015), and iSUN (Xu et al., 2015). To focus on the unknown rejection ability on distribution shifts, we first evaluate the performance with a mixture of covariate-shifted and semantic-shifted data at the inference stage and generally keep equal numbers of misclassified covariate-shifted data $\mathcal{P}_{\text{out}}^{\text{covariate}}(y \neq h(\mathbf{x}))$ and semantic OOD data $\mathcal{P}_{\text{out}}^{\text{semantic}}$, which are two kinds of failure sources we want to reject. Then, we provide the unified unknown rejection results evaluated on both clean ID and distribution-shifted data. We use the ResNet-18 (He et al., 2016) and optimize it with SGD optimizer for 200 epochs to get the standard pre-trained model. Then it is fine-tuned for 10 epochs to acquire different aspects of reliability. For LoRA, we simply set $r = 4$. More implementation details are provided in Appendix.

**Metrics and comparison methods.** We leverage AURC (‰) (Geifman & El-Yaniv, 2017; Jaeger et al., 2022), FPR95 (%) and AUC (%) (Hendrycks & Gimpel, 2016) to evaluate the performance of unknown rejection. Besides, we also introduce the F-AUC (%) (defined in Appendix). We compare TrustLoRA with various methods including CE (MSP) (Hendrycks & Gimpel, 2016), RegMixUp

Table 1: Unknown rejection performance under mixture of covariate and semantic shifts on CIFAR-10 with ResNet-18. Methods with * train from scratch, methods with + fully fine-tune the pretrained model, while others only fine-tune the LoRA.

| Method | Severity-1 | | | | Severity-2 | | | | Severity-3 | | | |
|---|---|---|---|---|---|---|---|---|---|---|---|---|
| | **AURC** | FPR95 | AUC | F-AUC | **AURC** | FPR95 | AUC | F-AUC | **AURC** | FPR95 | AUC | F-AUC |
| CE* | 56.04 | 37.53 | 89.49 | 87.41 | 89.05 | 44.23 | 86.93 | 84.19 | 122.75 | 48.93 | 84.63 | 81.17 |
| RegMixUp* | 57.05 | 50.60 | 88.56 | 86.86 | 89.13 | 55.62 | 86.30 | 83.89 | 124.50 | 58.52 | 84.05 | 80.99 |
| CRL* | 50.25 | 32.06 | 90.35 | 88.12 | 81.56 | 38.47 | 87.98 | 84.93 | 115.06 | 44.28 | 85.58 | 81.59 |
| LogitNorm* | 48.15 | 32.56 | 91.92 | 90.88 | 76.79 | 38.44 | 90.01 | 88.53 | 107.45 | 43.63 | 88.25 | 86.25 |
| OE* | 51.24 | 33.30 | 91.73 | 90.37 | 82.77 | 39.12 | 89.76 | 87.51 | 120.52 | 44.37 | 87.46 | 84.20 |
| OpenMix* | 29.46 | 28.13 | 92.45 | 91.08 | 46.90 | 32.61 | 90.95 | 89.12 | 66.86 | 36.94 | 89.18 | 86.83 |
| SURE* | 31.25 | 27.39 | 92.67 | 91.26 | 48.13 | 31.19 | 91.02 | 89.51 | 68.31 | 36.60 | 89.55 | 86.27 |
| RCL+ | 58.01 | 35.92 | 89.53 | 87.47 | 93.19 | 43.11 | 86.95 | 84.17 | 132.03 | 48.36 | 84.57 | 81.12 |
| SCONE+ | 44.01 | 26.99 | 93.13 | 92.08 | 71.28 | 32.75 | 91.17 | 89.37 | 104.06 | 38.91 | 88.83 | 86.15 |
| RegMixUp | 62.17 | 53.63 | 87.97 | 86.11 | 94.89 | 57.20 | 85.73 | 83.23 | 130.78 | 61.87 | 83.37 | 80.32 |
| CRL | 56.28 | 38.40 | 89.36 | 87.37 | 88.87 | 44.57 | 86.97 | 84.27 | 125.28 | 50.16 | 84.45 | 81.09 |
| LogitNorm | 59.05 | 36.71 | 89.41 | 87.22 | 94.15 | 42.89 | 86.84 | 83.85 | 133.41 | 48.76 | 84.26 | 80.62 |
| OE | 43.63 | 26.01 | 93.51 | 92.47 | 70.04 | 30.86 | 92.56 | 90.69 | 100.99 | 36.04 | 90.91 | 87.96 |
| AugMix | 36.62 | 36.09 | 90.72 | 89.16 | 51.73 | 39.06 | 89.61 | 87.71 | 67.82 | 41.97 | 88.51 | 86.20 |
| MaxLogit | 40.58 | 45.39 | 89.30 | 88.08 | 55.32 | 46.52 | 88.57 | 87.06 | 71.02 | 48.31 | 87.74 | 85.90 |
| Energy | 43.50 | 49.19 | 88.03 | 86.84 | 58.75 | 50.10 | 87.33 | 85.94 | 74.90 | 51.51 | 86.48 | 84.86 |
| KNN | 43.87 | 42.53 | 87.20 | 86.00 | 61.94 | 45.98 | 85.71 | 84.27 | 82.10 | 48.63 | 84.02 | 82.35 |
| FS-KNN | 47.54 | 55.75 | 87.29 | 86.01 | 62.36 | 55.35 | 86.79 | 85.30 | 80.36 | 57.86 | 85.52 | 83.74 |
| NNGuide | 51.77 | 63.95 | 85.25 | 84.05 | 68.17 | 63.28 | 84.65 | 83.32 | 84.80 | 63.49 | 84.01 | 82.40 |
| Relation | 58.98 | 59.52 | 80.85 | 79.74 | 77.10 | 60.19 | 80.17 | 78.85 | 97.12 | 61.66 | 79.04 | 77.57 |
| GEN | 41.30 | 46.00 | 88.78 | 87.62 | 56.34 | 47.31 | 88.00 | 86.61 | 72.20 | 48.94 | 87.18 | 85.48 |
| ASH | 41.10 | 45.87 | 89.11 | 87.89 | 55.97 | 47.28 | 88.31 | 86.83 | 72.08 | 49.31 | 87.44 | 85.66 |
| TrustLoRA | **28.68** | **23.80** | **93.67** | **92.53** | **41.64** | **27.43** | **92.67** | **91.18** | **56.64** | **30.62** | **91.55** | **89.65** |

(Pinto et al., 2022), CRL (Moon et al., 2020), LogitNorm (Wei et al., 2022), OE (Hendrycks et al., 2018), OpenMix (Zhu et al., 2023a), SURE (Li et al., 2024), RCL (Zhu et al., 2024a), AugMix (Hendrycks et al., 2019b), MaxLogit (Hendrycks et al., 2022), Energy (Liu et al., 2020), KNN (Sun et al., 2022), FS-KNN (Cen et al., 2023), NNGuide (Park et al., 2023), Relation (Kim et al., 2023a), GEN (Liu et al., 2023) and ASH (Djurisic et al., 2022). For training-time methods, we report the results of both training from scratch and LoRA fine-tuning. Score-based methods are applied to LoRA-augmented model tuning with AugMix. TrustLoRA leverages the simple MSP score (Hendrycks & Gimpel, 2016).

## 4.1 RESULTS AND DISCUSSION

To fully reflect the unknown rejection performance under both covariate and semantic shifts, we combine each of 15 corruptions under three different severity with six semantic OOD sets, resulting in 90 wild data mixtures in total. We report the average performance on those 90 evaluations.

**Trade-off between the two unknown rejection tasks.** Fig. 4 shows the performance change when fine-tuning the pre-trained model. Cov-MisD denotes the ability to reject misclassified covariate-shifted (e.g., `Gaussian noise`) examples. Unknown rejection denotes the ability to reject both misclassified covariate-shifted data and semantic-shifted data jointly. We can clearly observe that when fine-tuning with OE (after the dotted line) to acquire OOD detection ability, it is harder to detect misclassi-

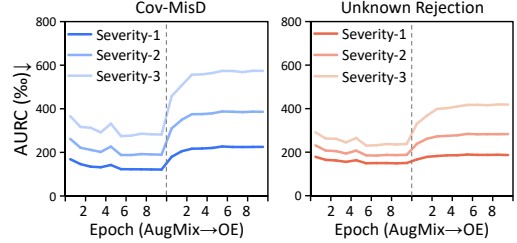

Figure 4: Change of rejection ability when fine-tuning the pre-trained ResNet-18 on CIFAR-100.

fied corrupted samples, e.g., AURC (↓) of Cov-MisD increases dramatically. As a result, the unified unknown rejection performance becomes worse.

**Our method achieves strong performance.** The main results in Table 1 and 2 verify that TrustLoRA establishes overall strong performance, especially on *AURC ↓, which has been considered as the most important metric for unknown rejection evaluation* (Jaeger et al., 2022; Moon et al., 2020). In particular, we consider two groups of baselines: training the model from scratch (denotes with *) and fine-tuning the pre-trained model. We highlight a few observations: (1) TrustLoRA

Table 2: Unknown rejection under mixture of covariate and semantic shifts on CIFAR-100.

| Method | Severity-1 | | | | Severity-2 | | | | Severity-3 | | | |
|---|---|---|---|---|---|---|---|---|---|---|---|---|
| | **AURC** | FPR95 | AUC | F-AUC | **AURC** | FPR95 | AUC | F-AUC | **AURC** | FPR95 | AUC | F-AUC |
| CE* | 163.66 | 54.88 | 82.95 | 76.33 | 218.26 | 61.90 | 79.53 | 71.53 | 279.10 | 67.12 | 76.65 | 68.06 |
| RegMixUp* | 154.87 | 54.34 | 83.12 | 76.72 | 198.08 | 60.46 | 80.25 | 72.64 | 249.93 | 65.33 | 77.43 | 69.25 |
| CRL* | 152.20 | 53.03 | 83.94 | 77.67 | 195.12 | 59.43 | 81.07 | 73.37 | 244.45 | 64.10 | 78.74 | 70.00 |
| LogitNorm* | 166.34 | 56.51 | 82.91 | 77.24 | 217.56 | 62.15 | 80.21 | 73.37 | 273.87 | 66.38 | 77.78 | 70.37 |
| OE* | 149.02 | 45.48 | 87.09 | 83.17 | 194.97 | 51.69 | 85.26 | 80.21 | 246.45 | 56.24 | 83.31 | 77.18 |
| OpenMix* | 134.18 | 46.46 | 86.57 | 81.84 | 164.32 | 51.49 | 84.47 | 78.78 | 203.87 | 56.18 | 82.29 | 76.10 |
| SURE* | 137.62 | 48.35 | 86.04 | 82.26 | 172.45 | 51.76 | 83.95 | 78.41 | 205.71 | 56.62 | 82.18 | 76.54 |
| RCL+ | 155.85 | 52.89 | 84.29 | 78.73 | 202.39 | 59.20 | 81.61 | 74.88 | 256.47 | 64.10 | 79.11 | 71.83 |
| SCONE+ | 148.50 | 47.65 | 86.50 | 81.69 | 201.73 | 53.92 | 83.57 | 77.37 | 264.59 | 59.29 | 80.74 | 74.03 |
| RegMixUp | 155.58 | 53.51 | 83.81 | 77.99 | 203.97 | 59.66 | 80.70 | 73.83 | 261.76 | 64.50 | 77.96 | 70.71 |
| CRL | 153.48 | 52.10 | 84.27 | 78.57 | 204.13 | 58.43 | 81.24 | 74.25 | 262.74 | 63.65 | 78.31 | 71.19 |
| LogitNorm | 155.08 | 52.55 | 84.14 | 78.38 | 207.23 | 58.92 | 81.07 | 74.10 | 267.35 | 63.96 | 78.16 | 71.06 |
| OE | 147.40 | 46.57 | 87.21 | 83.04 | 197.74 | 52.09 | 84.73 | 79.32 | 259.43 | 57.08 | 82.12 | 76.15 |
| AugMix | 141.23 | 51.24 | 84.69 | 79.69 | 158.07 | 54.44 | 83.47 | 77.63 | 177.71 | 57.12 | 82.26 | 75.69 |
| MaxLogit | 150.37 | 57.84 | 83.33 | 78.72 | 166.31 | 59.89 | 82.35 | 76.96 | 185.83 | 61.91 | 81.20 | 75.10 |
| Energy | 158.37 | 60.62 | 81.52 | 77.58 | 175.02 | 62.48 | 80.47 | 75.90 | 194.12 | 64.27 | 79.40 | 74.28 |
| KNN | 168.58 | 66.07 | 80.55 | 77.54 | 184.77 | 67.07 | 79.41 | 75.79 | 204.44 | 68.17 | 78.22 | 74.09 |
| FS-KNN | 143.83 | 56.08 | 84.71 | 81.69 | 167.36 | 58.13 | 83.03 | 79.74 | 183.17 | 59.21 | 82.26 | 77.12 |
| NNGuide | 180.32 | 66.12 | 76.27 | 72.23 | 197.55 | 67.51 | 75.21 | 70.47 | 215.89 | 68.58 | 74.39 | 69.10 |
| Relation | 171.64 | 67.49 | 80.28 | 76.20 | 186.00 | 67.95 | 79.49 | 74.63 | 204.07 | 68.98 | 78.58 | 73.07 |
| GEN | 157.47 | 60.25 | 81.68 | 77.69 | 174.32 | 62.19 | 80.57 | 75.99 | 193.63 | 64.04 | 79.50 | 74.32 |
| ASH | 150.38 | 57.78 | 83.22 | 78.77 | 167.33 | 59.93 | 82.09 | 76.90 | 185.92 | 61.75 | 81.05 | 75.13 |
| TrustLoRA | **129.64** | **46.46** | **87.29** | **83.73** | **149.14** | **50.12** | **85.77** | **81.40** | **172.35** | **53.32** | **84.41** | **79.28** |

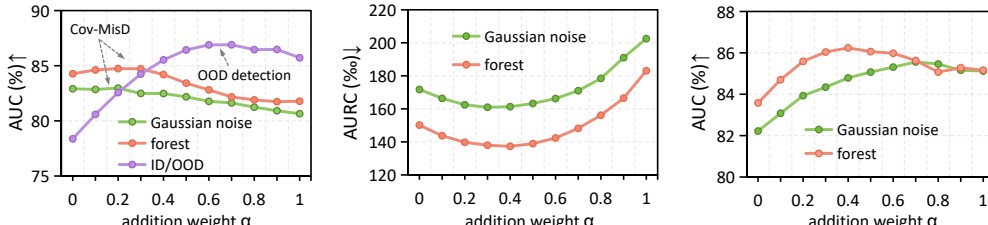

Figure 5: Flexibility of controlling the strength of reliability edition on CIFAR-100.

outperforms strong training methods like LogitNorm (Wei et al., 2022), CRL (Moon et al., 2020) and RegMixUp (Pinto et al., 2022) in both training from scratch and fine-tuning scenarios. (2) TrustLoRA outperforms competitive post-hoc OOD detection methods, which are applied to the same model fine-tuned with AugMix and hence they have the same classification accuracy. (3) The proposed reliability arithmetic framework excels in detecting both misclassified covariate-shifted and semantic-shifted data, achieving the best performance among all compared methods.

**Flexibility of controlling the strength of reliability edition.** We separate reliability knowledge with LoRAs and merge them to get a unified failure detector. One of the primary advantages of our method is to control the strength of each kind of reliability flexibly based on end-user preference without training the model again or affecting the original model. In Fig. 5 (Left), we show that the scaling $\alpha$ in Eq. (7) can easily control the preference between MisD under covariate shits and OOD detection. In Fig. 5 (Middle and Right), we observe that an overall

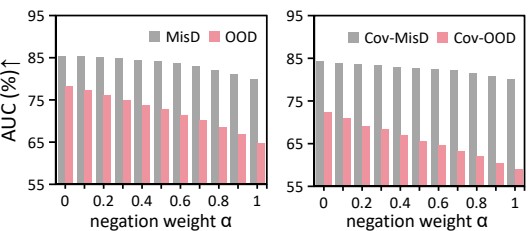

Figure 6: Accurate forgetting of OOD detection ability while keeping the MisD ability on clean ID (Left) and covariate-shifted data (Right).

strong unified unknown rejection performance can be achieved with $\alpha \in [0.4, 0.6]$, and we simply set $\alpha = 0.5$ for all experiments.

**Selective reliability forgetting with LoRA negation.** Besides LoRA addition for unified unknown rejection, here we explore accurate reliability forgetting. We apply the OOD detection vector $\tau = -\alpha \cdot \tau_{\text{LoRA,sem}}$ (learned with OE) to a given model (e.g., LoRA tuning with AugMix). The experiments on CIFAR-100 in Fig. 6 show that we can enable the model to forget OOD detection

Table 3: Unknown rejection performance on CIFAR-100 with ViT.

| Method | Severity-1 | | | | Severity-2 | | | | Severity-3 | | | |
|---|---|---|---|---|---|---|---|---|---|---|---|---|
| | **AURC** | **FPR95** | **AUC** | **F-AUC** | **AURC** | **FPR95** | **AUC** | **F-AUC** | **AURC** | **FPR95** | **AUC** | **F-AUC** |
| Full-FT | 49.17 | 33.80 | 91.07 | 89.09 | 69.78 | 37.67 | 89.77 | 87.08 | 92.46 | 41.43 | 88.39 | 84.81 |
| Linear | 98.88 | 37.34 | 90.03 | 87.23 | 116.38 | 40.22 | 88.94 | 85.33 | 134.55 | 43.36 | 87.70 | 83.20 |
| CE | 51.68 | 33.77 | 91.30 | 89.61 | 73.33 | 36.91 | 90.14 | 87.88 | 93.37 | 40.73 | 88.87 | 85.95 |
| RegMixUp | 50.54 | 37.70 | 90.83 | 89.21 | 74.42 | 44.21 | 88.97 | 86.78 | 97.86 | 49.19 | 87.35 | 84.41 |
| CRL | 54.48 | 34.01 | 91.23 | 89.44 | 76.32 | 37.63 | 90.01 | 87.65 | 95.83 | 40.86 | 88.74 | 85.70 |
| LogitNorm | 59.66 | 49.24 | 88.70 | 87.39 | 81.65 | 50.18 | 88.16 | 86.44 | 101.24 | 51.96 | 87.25 | 85.03 |
| AugMix | 46.13 | 34.27 | 91.12 | 89.37 | 65.15 | 37.65 | 90.06 | 87.75 | 85.36 | 41.12 | 88.83 | 85.89 |
| MaxLogit | 52.22 | 43.66 | 89.35 | 88.08 | 70.65 | 45.35 | 88.76 | 87.11 | 89.95 | 47.21 | 87.93 | 85.79 |
| Energy | 55.52 | 48.50 | 88.24 | 87.00 | 74.24 | 49.68 | 87.76 | 86.18 | 93.74 | 51.18 | 87.02 | 85.07 |
| KNN | 58.91 | 53.46 | 86.82 | 85.46 | 78.90 | 54.02 | 86.11 | 84.51 | 99.20 | 54.78 | 85.32 | 83.38 |
| NNGuide | 53.86 | 46.97 | 88.53 | 87.28 | 72.53 | 48.36 | 87.96 | 86.40 | 92.06 | 50.20 | 87.17 | 85.23 |
| GEN | 54.15 | 46.31 | 88.59 | 87.36 | 73.01 | 47.82 | 88.00 | 86.48 | 92.40 | 49.49 | 87.25 | 85.28 |
| TrustLoRA | **43.40** | **31.25** | **92.13** | **90.78** | **63.01** | **34.08** | **91.22** | **89.36** | **83.37** | **37.91** | **89.90** | **87.56** |

ability, while with little deterioration of MisD ability on clean ID (Left) and covariate-shifted data (Right). This demonstrates that our method can enable flexible reliability knowledge edition.

**Experiments with ViT.** We also conduct experiments on pre-trained ViT backbone (ViT-B16) (Dosovitskiy et al., 2020), and perform full fine-tuning, linear prob and LoRA tuning. Detailed implementation can be found in Appendix. Despite the strong performance of pre-trained ViT-B16, results in Table 3 reveal that our method yields notable improvement, especially on AURC.

**Large scale experiments on ImageNet.** We provide additional large-scale results on the ImageNet-200/500 benchmark with ResNet-50. The classes were randomly sampled from 1K, and we also sampled another set of classes (with equal numbers) as outliers for OE. At inference stage, we use a mixture of covariate and semantic OOD data. Specifically, for semantic shifts, we use the fixed ImageNet OOD dataset proposed in (Bitterwolf et al., 2023), which includes truly OOD versions of 11 popular OOD datasets with in total of 2715 OOD samples; for covari-

Table 4: Experimental results on ImageNet.

| Method | ImageNet-200 | | ImageNet-500 | |
|---|---|---|---|---|
| | **AURC** | **AUC** | **AURC** | **AUC** |
| CE* | 188.37 | 92.55 | 268.42 | 89.08 |
| MaxLogit* | 198.44 | 90.11 | 286.90 | 85.03 |
| Energy* | 203.02 | 89.28 | 295.91 | 83.71 |
| AugMix (LoRA) | 166.53 | 93.25 | 236.25 | 89.95 |
| OE (LoRA) | 180.18 | 92.78 | 265.11 | 89.50 |
| MaxLogit (AugMixLoRA) | 187.54 | 91.97 | 259.65 | 87.82 |
| Energy (AugMixLoRA) | 197.60 | 91.63 | 266.37 | 86.65 |
| TrustLoRA | **159.67** | **93.91** | **229.92** | **90.15** |

ate shifts, we use the corruption type `Frost` with severity-1. Results in Table 4 suggest that our method yields strong unknown rejection performance compared with competitive baselines.

**TrustLoRA outperforms the multi-task learning.** We further compare our method with more baselines: (1) Two new methods named SIRC (Xia & Bouganis, 2024) and FMFP (Zhu et al., 2023b) (FlatLoRA in our comparison). (2) Multi-task tuning with combined OE and Aug-Mix learning objectives. The results in Table 5 verify that our method outperforms SIRC and FlatLoRA consistently. In particular, our LoRA arithmetic outperforms the multi-task learning,

Table 5: Comparison with more baselines and multi-task learning on CIFAR-100, severity-1.

| Method | **AURC** | **FPR95** | **AUC** | **F-AUC** |
|---|---|---|---|---|
| SIRC* (MSP, $z1$) | 160.26 | 52.17 | 83.75 | 76.65 |
| FlatLoRA | 152.94 | 51.97 | 84.40 | 78.74 |
| SIRC (AugMixLoRA) | 139.86 | 50.53 | 85.50 | 80.04 |
| AugMix (Full FT) | 133.68 | 49.79 | 85.05 | 80.48 |
| AugMix+OE (Full FT) | 138.92 | 50.26 | 85.40 | 81.44 |
| TrustLoRA | **129.64** | **46.46** | **87.29** | **83.73** |

i.e., AugMix+OE (Full FT) in Table 5. *Intuitively*, this is because when optimizing both two objectives in a multi-task learning (MTL) manner, there exist remarkable conflicts between pulling covariate-shifted samples close to class centers while pushing semantic-shifted samples away from class centers since those two types of shifted samples are often overlapping. *Theoretically*, the Bayes-optimal reject rule for MisD is based on maximum class-posterior probability $\max_{y \in \mathcal{Y}} \mathbb{P}(y|\mathbf{x})$, while OOD detection rejects samples with small density ratio $p(\mathbf{x}|\text{in})/p(\mathbf{x}|\text{out})$ (Zhu et al., 2023b; Narasimhan et al., 2024). OOD detection methods such as OE and Energy score often perform density estimation explicitly or implicitly. However, to separate samples from known classes and unknown semantic-shifted unknown classes, binary discrimination would compress the confidence distribution of correct and incorrect covariate-shifted samples. As a result, MTL of the two objectives would be unstable. Differently, our proposed LoRA arithmetic overcomes the above limitation with reliability knowledge separation and consolidation.

Table 6: Robustness to different auxiliary data when acquiring OOD detection ability.

| Auxiliary Data | Severity-1 | | | | Severity-2 | | | | Severity-3 | | | |
|---|---|---|---|---|---|---|---|---|---|---|---|---|
| | AURC | FPR95 | AUC | F-AUC | AURC | FPR95 | AUC | F-AUC | AURC | FPR95 | AUC | F-AUC |
| TIN597 | 130.45 | 45.26 | 87.41 | 83.19 | 152.50 | 49.63 | 85.81 | 80.83 | 178.82 | 52.73 | 84.30 | 78.66 |
| RandomImage | 129.64 | 46.46 | 87.29 | 83.73 | 149.14 | 50.12 | 85.77 | 81.40 | 172.35 | 53.32 | 84.41 | 79.28 |

**Robustness to different auxiliary data.** In this paper, the proposed TrustLoRA acquires OOD detection via OE technique, which requires access to auxiliary outlier data. For CIFAR benchmark, the `RandomImage` is used as auxiliary outliers following existing work. In Table 6 (CIFAR-100), we show that TrustLoRA is robust to other auxiliary outliers like TIN597 (Zhang et al., 2023b).

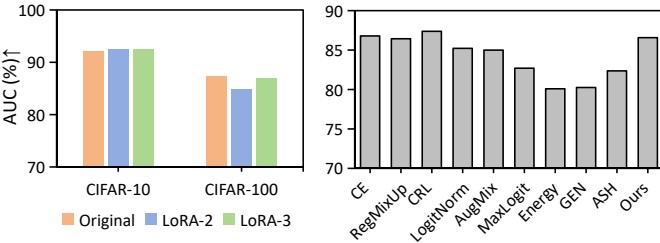

Figure 7: MisD ability on clean ID.

**Unified unknown rejection on clean ID and distribution-shifted data.** In above experiments, to *clearly reflect the unknown rejection ability under covariate and semantic shits*, we do not include clean ID data at inference time. With integrated LoRAs of covariate and semantic shifts, the MisD performance on the original clean set can be well preserved on CIFAR-10 while suffering from a slight drop on CIFAR-100. As shown in Fig. 7 (Left), TrustLoRA can further recover and integrate

Table 7: Comparison of unified unknown rejection ability evaluated on both clean ID and distribution-shifted data.

| Method | CIFAR-10 | | | | CIFAR-100 | | | |
|---|---|---|---|---|---|---|---|---|
| | AURC | FPR95 | AUC | F-AUC | AURC | FPR95 | AUC | F-AUC |
| CE | 32.62 | 31.56 | 91.59 | 89.98 | 134.19 | 48.70 | 85.47 | 79.71 |
| RegMixUp | 34.82 | 44.92 | 90.56 | 89.27 | 133.88 | 49.25 | 84.96 | 79.35 |
| CRL | 30.37 | 26.24 | 92.30 | 90.54 | 129.75 | 47.73 | 85.97 | 80.47 |
| LogitNorm | 27.11 | 26.10 | 93.76 | 92.97 | 138.73 | 51.38 | 85.22 | 80.20 |
| AugMix | 27.47 | 31.75 | 91.85 | 90.45 | 133.73 | 49.50 | 85.34 | 80.70 |
| MaxLogit | 32.53 | 46.17 | 89.95 | 88.86 | 142.31 | 56.11 | 84.00 | 79.74 |
| Energy | 35.48 | 50.48 | 88.52 | 87.49 | 149.95 | 58.96 | 82.23 | 78.55 |
| GEN | 33.33 | 46.73 | 89.31 | 88.31 | 149.52 | 58.66 | 82.30 | 78.59 |
| ASH | 33.09 | 46.95 | 89.67 | 88.61 | 142.67 | 56.05 | 83.87 | 79.75 |
| TrustLoRA | **20.86** | **20.86** | **94.42** | **93.47** | **121.90** | **44.44** | **87.75** | **84.35** |

MisD knowledge on clean set by merging an additional LoRA fine-tuned with flat minima loss Zhu et al. (2023b), and we denote the model "LoRA-3". Fig.7 (Right) compares the MisD on clean ID data, where our method successfully achieves comparable MisD performance with the original model, and outperforms other methods. Table 7 further reports the results on full spectrum of test set including clean ID, covariate and semantic OOD data. As can be observed, our method still achieves strong performance and outperforms other methods.

**Computational costs.** Table 8 reports the number of parameters of the base model and all LoRA modules, where our method has much smaller parameters than the base model. Note that we further fix the parameters of **A** once initialized and only optimize **B** in LoRA during the training stage, which is much more efficient than learning the original LoRA.

Table 8: Comparison of the computational costs.

| Model | ResNet-20 | ResNet-18 | ViT (B16) |
|---|---|---|---|
| BaseModel | $0.2871M$ | $10.91M$ | $81.89M$ |
| TrustLoRA | $\mathbf{0.0275}M$ | $\mathbf{0.25}M$ | $\mathbf{0.21}M$ |

## 5 CONCLUSION

In this work, we present a novel reliability arithmetic framework to address the unknown rejection under both covariate and semantic shifts. For the first time, we introduce low-rank adaptation to separate and compress reliability knowledge. The proposed framework is a powerful tool to easily achieve unified, flexible and controllable reliability towards different failure sources. Extensive experiments and analysis show the superiority of our method over existing approaches for unknown rejection under both covariate and semantic shifts. We hope this work can inspire the community to investigate the trade-off among different failure sources, and further develop flexible and controllable methods for reliable prediction in real-world applications.

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

## A  RELATED WORK

**Covariate OOD generalization.** To improve the generalization, some methods assume that a set of covariate-shifted samples are available at training time (Yao et al., 2022; Wang et al., 2022; Shi et al., 2021; Bai et al., 2023), others aim to learn domain-invariant representations that generalize better under covariate shifts (Rusak et al., 2020; Hendrycks et al., 2019b;a). Recently, Bai et al., (Bai et al., 2023) leveraged unlabeled wild data consisting of covariate and semantic shifts to build a model to recognize covariate-shifted data while rejecting semantic-shifted data. However, there is no rejection option in covariate shits generalization, and users would accept the widely existing misclassification blindly. Contrary to the above prior works, in this work, we aim to reject misclassified covariate-shifted samples reliably.

**Semantic OOD detection.** Current OOD detection methods have been proposed under the setting of post hoc or training regularization, aiming to reject samples from unseen classes. Some post hoc methods (Hendrycks & Gimpel, 2016; Liu et al., 2020; Liang et al., 2018; Basart et al., 2022; Huang et al., 2021; Sun et al., 2022; Hendrycks et al., 2022; Park et al., 2023; Kim et al., 2023a; Liu et al., 2023) focus on designing proper confidence scores, which others (Sun et al., 2021; Sun & Li, 2022; Djurisic et al., 2022; Song et al., 2022; Djurisic et al., 2022) remove undesirable parts of feature or activation to facilitate the separation of ID and OOD examples. Training regularization approaches (Hendrycks et al., 2018; Du et al., 2024; Ming et al., 2022; Zheng et al., 2024; Du et al., 2021; Zhu et al., 2024b; Zhang et al., 2023a; Katz-Samuels et al., 2022) often require real or synthesized auxiliary dataset with extra training processes. Nevertheless, current OOD detection methods could harm the performance of detecting misclassified examples from known classes. This work aims to develop unified and flexible framework to detect different kinds of failures.

Recently, there are a few studies (Zhu et al., 2024a; Cen et al., 2023; Zhu et al., 2023a; Li et al., 2024) focused on developing reliable models that can reject both misclassified ID and semantic-shifted OOD data. For example, Zhu et al. (2023a; 2024a) observed that existing popular OOD detection methods are harmful for misclassification detection on clean ID test data, and proposed unified failure detection methods by exploring outlier data (Zhu et al., 2023a) or reliable continual learning paradigm (Zhu et al., 2024a). Cen et al. (2023) found that the uncertainty distribution of wrongly classified samples is extremely close to semantic-shifted samples rather than known and correctly classified samples, and proposed FS-KNN, which is an improvement of the KNN score. Li et al. (2024) proposed a method named SURE for reliable prediction by combining multiple techniques, across model regularization, classifier and optimization. Nevertheless, they typically overlook covariate-shifted samples, and it is hard to distinguish correct covariate-shifted samples from semantic-shifted ones. Besides, they often train a model from scratch or fully fine-tune it, which is computationally heavy and inefficient.

## B  THEORETICAL ANALYSIS

### B.1  PROOF OF PROPOSITION 3.1

**Proposition:** Given a compact $\mathcal{X} \subseteq \mathbb{R}^D$ and a family of continuous mappings $f_n : \mathcal{X} \to \mathbb{R}^{D'}$, $n = 1, \ldots, N$, there exists a ReLU multi-layer perceptron $f$ with base parameters $\theta_{\text{pre}}$ and two low-rank vectors $\tau_{\text{LoRA,cov}}$ and $\tau_{\text{LoRA,sem}}$, such that for any $\epsilon > 0$ and all $n$, there exists an $\alpha \in [0, 1]$ satisfying

$$\|f_n(\boldsymbol{x}) - f(\boldsymbol{x}; \theta_{\text{pre}} + (1 - \alpha) \cdot \tau_{\text{LoRA,cov}} + \alpha \cdot \tau_{\text{LoRA,sem}})\| \leq \epsilon, \quad \forall \boldsymbol{x} \in \mathcal{X}.$$

**Proof:** The proof is based on (Dimitriadis et al., 2023). Formally, denote $\sigma$ the ReLU non-linearity $\sigma(x) = \max(0, x)$. From the universal approximation theorem (Haykin, 1998), for any $\epsilon > 0$, there exists $Q \in \mathbb{N}$, $M \in \mathbb{R}^{(D+2) \times Q}$, $C \in \mathbb{R}^Q$, $M' \in \mathbb{R}^{Q \times D'}$ such that:

$$\forall \boldsymbol{x} \in \mathcal{X}, \forall n \in \{1, \ldots, N\}, \quad \|f_n(\boldsymbol{x}) - g(\boldsymbol{x}, \alpha)\| \leq \epsilon,$$

where $g(\boldsymbol{x}, \alpha) = M' \sigma(M(\boldsymbol{x}, \alpha) + C)$.

Define two matrices $R \in \mathbb{R}^{D \times (2D+2)}$ and $S \in \mathbb{R}^{(2D+2) \times D}$ as follows:

$$R_{i,j} = \begin{cases} 1, & \text{if } j = 2i - 1, \\ -1, & \text{if } j = 2i, \\ 0, & \text{otherwise}, \end{cases} \quad \text{and} \quad S_{i,j} = \begin{cases} 1, & \text{if } i = 2j - 1 \text{ or } (i > 2u \text{ and } j = u), \\ -1, & \text{if } i = 2j, \\ 0, & \text{otherwise}. \end{cases}$$

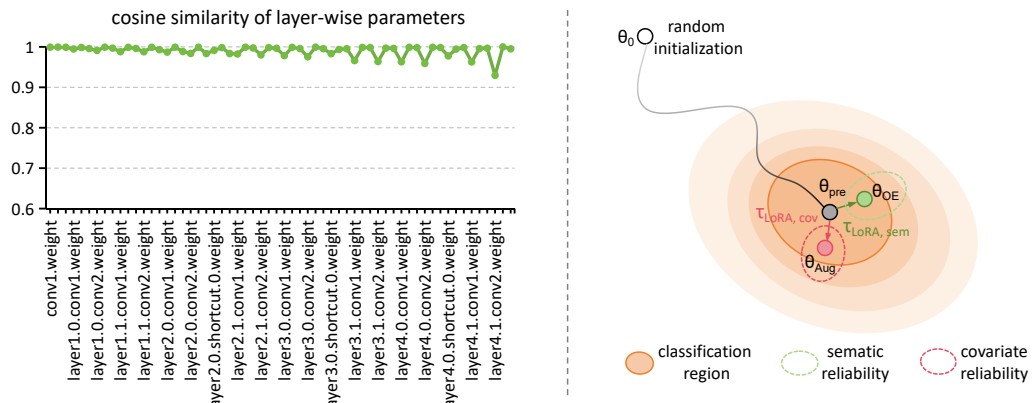

Figure 8: (Left) Discrepancy (measured by layer-wise cosine similarity) between the pre-trained model and fine-tuned model with OE. (Right) Illustration of the distribution of classification region, covariate and semantic reliability region.

Let $W_k = (0, \ldots, 0, k), k = 1, 2$. Then, with $\boldsymbol{x} \in \mathbb{R}^D$, we have:

$$\forall \alpha \geq 0, \quad S^\top \sigma(R^\top \boldsymbol{x} + (1-\alpha)W_1 + \alpha W_2) = (\boldsymbol{x}, \alpha).$$

We can learn a ReLU multi-layer perceptron $f(\boldsymbol{x}, M, C, M', R, S, W) = M'\sigma(MS^\top \sigma(R^\top \boldsymbol{x} + W) + C)$. Then with $\theta_{\text{pre}} = (M, C, M', R, S, 0)$ and $\theta_i = (0, 0, 0, 0, 0, W_i)$ for $i = 1, 2$, we have:

$$\begin{aligned}
f(\boldsymbol{x}; \theta_{\text{pre}} + (1-\alpha) \cdot \theta_1 + \alpha \cdot \theta_2) &= f(\boldsymbol{x}; M, C, M', R, S, (1-\alpha)W_1 + \alpha W_2) \\
&= M'\sigma(MS^\top \sigma(R^\top \boldsymbol{x} + (1-\alpha)W_1 + \alpha W_2) + C) \\
&= g(S^\top(R^\top \boldsymbol{x} + (1-\alpha)W_1 + \alpha W_2)) \\
&= g(\boldsymbol{x}, \alpha).
\end{aligned}$$

Note that $\theta_i, i = 1, 2$ can be reshaped into a matrix $B_i A_i$, and in this paper we define them as $\theta_1 = \tau_{\text{LoRA,cov}}$ and $\theta_2 = \tau_{\text{LoRA,sem}}$, respectively.

**Remark.** Recently, Zeng & Lee (2024) has studied the expressive power of LoRA, providing several conditions for LoRA to be an exactly universal approximator. When the rank of LoRA is lower than the critical threshold, the authors provided an upper bound for the approximation error. Specifically, the approximation error is related to i) the magnitude of the target model's parameters and the input; ii) the rank of the adapter and the discrepancy between the frozen model and the target model; iii) the depth of the frozen model. In our work, we do not focus on an exact universal approximator. The low-rank module is used to approximate the residual parameters between the pre-trained model and the failure-specific fine-tuned model. As shown in Fig. 8, the similarities of layer-wise parameters between the pre-trained model (ResNet-18, CIFAR-100) and the fine-tuned model are very high. Therefore, the parameter discrepancy is small. Our proof is based on the universal approximation theorem with unconstrained width of the LoRA module, and the $\epsilon$ expresses the approximation error, which shares a similar spirit with that in (Zeng & Lee, 2024).

## C    EXPERIMENTS

### C.1    EXPERIMENTAL SETUP DETAILS

For pre-trained model, we train the ResNet-18 model with SGD optimizer, a momentum of 0.9, an initial learning rate of 0.1, a weight decay of 5e-4 and mini-batch size of 128. The number of training epoch is 200, and the learning rate is reduced by a factor of 10 at 100, and 150 epochs.

Training configures for Augmix: For augmentation, we use the official AugMix code and follow the setup on the original AugMix paper to randomly sample $k$ augmentation chains, where $k = 3$ by default. The sample mixing weights $(w1, w2, ..., wk) \sim \text{Dirichlet}(\alpha, \alpha, ..., \alpha)$, where $\alpha = 1$ by default. The $\lambda$ is set to be 12 following the official code of AugMix at line 234 in the above address.

Table 9: Unknown rejection ability under covariate and semantic shits can be well maintained.

| Method | Severity-1 | | | | Severity-2 | | | | Severity-3 | | | |
|---|---|---|---|---|---|---|---|---|---|---|---|---|
| | AURC | FPR95 | AUC | F-AUC | AURC | FPR95 | AUC | F-AUC | AURC | FPR95 | AUC | F-AUC |
| TrustLoRA (LoRA-2) | 129.64 | 46.46 | 87.29 | 83.73 | 149.14 | 50.12 | 85.77 | 81.40 | 172.35 | 53.32 | 84.41 | 79.28 |
| TrustLoRA (LoRA-3) | 128.14 | 46.65 | 86.33 | 81.68 | 140.64 | 49.45 | 85.29 | 80.00 | 155.53 | 52.04 | 84.25 | 78.29 |

Table 10: Results on less overparameterised model.

| Method | MD-AURC | MD-FPR | MD-AUC | OOD-FPR | OOD-AUC | UR-AURC | UR-FPR | UR-AUC | F-AUC |
|---|---|---|---|---|---|---|---|---|---|
| Baseline (C10) | 57.28 | 41.60 | 86.93 | 48.24 | 81.47 | 85.80 | 40.92 | 87.05 | 84.11 |
| w/ TrustLoRA | **41.89** | **38.46** | **87.99** | **41.92** | **84.64** | **69.78** | **36.79** | **88.81** | **86.28** |
| Baseline (C100) | 210.34 | 54.94 | 81.61 | 84.31 | 62.14 | 222.40 | 67.65 | 78.30 | 70.56 |
| w/ TrustLoRA | **165.16** | **51.72** | **83.14** | **78.52** | **67.43** | **188.41** | **62.10** | **81.08** | **74.46** |

We then train the ResNet-18 model with SGD optimizer, a momentum of 0.9, an initial learning rate of 0.1, a weight decay of 5e-4, and a mini-batch size of 128. The number of training epochs is 200, and the learning rate is reduced by a factor of 10 at 100, and 150 epochs.

Training configures for TrustLoRA: For AugMixLoRA fine-tuning, we set the rank of the LoRA as 4 and use a cosine learning rate with an initial learning rate of 0.001 and a total 10 epochs. The augmentation configures are the same as that of AugMix described above.

For experiments on ViT, we use the pre-trained ViT-B16, which is fine-tuned for 10 epochs using cosine learning rate with the initial learning rate of 0.03. We set the momentum to be 0.9 and the weight decay to 0. For compared methods, the main hyper-parameters come from their original papers. For KNN, we set $k$ to 50. For NNGuide, we set $k$ to 100. For GEN, the parameters $(\gamma, M)$ in calculating generalized entropy score are set to $(0.1, 100)$. We run each trial 3 times and report the average performance.

For evaluation metric, the AURC, FPR95 and AUC are widely used in prior works. We further define the F-AUC as follow: $F - AUC = (2 \times AUC_{cov} \times AUC_{sem})/(AUC_{cov} + AUC_{sem})$, where $AUC_{cov}$ denotes the AUC value of separating correct and incorrect covariate-shifted data and $AUC_{cov}$ denotes the AUC value of separating covariate-shifted and semantic-shifted data.

## C.2 ADDITIONAL RESULTS

**Unknown rejection performance can be well maintained.** In Table 9, we show that the unknown rejection ability under covariate and semantic shits can be well maintained after further integrating the third LoRA, which demonstrates the flexibility and effectiveness of the proposed TrustLoRA framework.

**Results on less overparameterised model.** We conduct experiments on ResNet-20 (which is much small than ResNet-18) for CIFAR-10/100 and the fine-grained results are shown in Table 10. As can be seen, the proposed Trust-LoRA successfully enhances the misclassification detection, OOD detection, and unknown rejection ability of the base model (0.287M) by only tuning a very small number (0.0275M) of parameters in LoRAs. Those results verify that our method can capture the benefit of Aug-Mix/OE when models are less overparameterised.

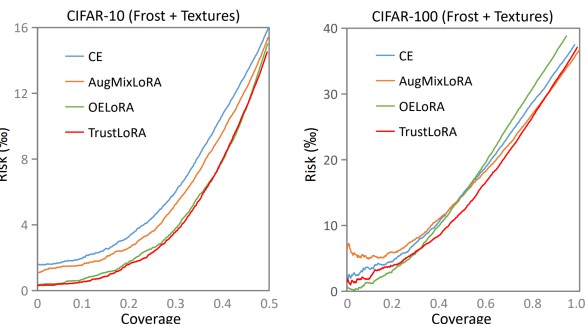

Figure 9: Risk-coverage curves on the mixture of ID, covariate and semantic shifts.

**More results in comparison with more baselines and multi-task learning.** Table 11 provides more results of comparison with SIRC, FlatLoRA and multi-task learning (i.e., AugMix+OE (Full FT)).

Table 11: Comparison with more baselines and multi-task learning on CIFAR-100, severity-1.

| Method | Severity-1 | | | | Severity-2 | | | |
|---|---|---|---|---|---|---|---|---|
| | **AURC** | FPR95 | AUC | F-AUC | **AURC** | FPR95 | AUC | F-AUC |
| SIRC* (MSP,z1) | 160.26 | 52.17 | 83.75 | 76.65 | 214.24 | 60.07 | 80.22 | 71.85 |
| FlatLoRA | 152.94 | 51.97 | 84.40 | 78.74 | 201.94 | 58.51 | 81.43 | 74.46 |
| SIRC (AugMixLoRA) | 139.86 | 50.53 | 85.50 | 80.04 | 156.63 | 54.03 | 83.86 | 78.25 |
| AugMix (Full FT) | 133.68 | 49.79 | 85.05 | 80.48 | 146.83 | 52.47 | 83.92 | 78.66 |
| AugMix+OE (Full FT) | 138.92 | 50.26 | 85.40 | 81.44 | 142.66 | 50.66 | 84.52 | 80.12 |
| TrustLoRA | **129.64** | **46.46** | **87.29** | **83.73** | **149.14** | **50.12** | **85.77** | **81.4** |

Table 12: Random projection based LoRA *v.s.* the original LoRA on CIFAR-10/100, severity-1.

| Method | CIFAR-10 | | | | CIFAR-100 | | | |
|---|---|---|---|---|---|---|---|---|
| | **AURC** | FPR95 | AUC | F-AUC | **AURC** | FPR95 | AUC | F-AUC |
| B only | 28.68 | 23.80 | 93.67 | 92.53 | 129.64 | 46.46 | 87.29 | 83.73 |
| A & B | 28.07 | 24.12 | 93.82 | 92.95 | 127.51 | 45.72 | 87.17 | 83.34 |

Table 13: Performance of TrustLoRA with different rank $r$ on CIFAR-10/100, severity-1.

| Method | CIFAR-10 | | | | CIFAR-100 | | | |
|---|---|---|---|---|---|---|---|---|
| | **AURC** | FPR95 | AUC | F-AUC | **AURC** | FPR95 | AUC | F-AUC |
| $r = 2$ | 29.31 | 24.92 | 93.32 | 92.24 | 131.06 | 47.17 | 87.11 | 83.38 |
| $r = 4$ | 28.68 | 23.80 | 93.67 | 92.53 | 129.64 | 46.46 | 87.29 | 83.73 |
| $r = 8$ | 28.45 | 23.11 | 93.60 | 92.72 | 128.15 | 46.03 | 87.40 | 83.82 |

**Detailed individual unknown rejection performance.** Table 14 provides the fine-grained results for misclassification detection on covariate shifts, OOD detection on semantic shifts and the unified unknown rejection results on C10/100 with severity-1. We observe that our method achieves the best unknown rejection performance. Besides, it is worth mentioning that our formulation is different from existing works that evaluate OOD generalization via accuracy and OOD detection via rejection metrics like FPR95, and AUC. We use the ARUC metric to reflect the classification with rejection ability on covariate shifts, which has integrated the classification and rejection performance. We also added the accuracy performance in Table 14, including methods that train the model from scratch (*) and others that fine-tune the model with LoRA. All post hoc methods have the same accuracy as AugMixLoRA, since they are applied to the model trained with AugMixLoRA.

**Risk-coverage curves.** Fig. 9 provides the comparison of the risk-coverage curves when testing on the mixture of covariate shifts (Frost with Severity-1) and semantic shifts (Textures) on CIFAR10 and CIFAR-100 datasets. Ours achieves the smallest risk given a specific coverage value.

**Compare the random projection based LoRA with the original LoRA.** In our method, we propose LoRA with random projection, where only the $B$ matrix of LoRA is trained. In Table 12, we show that random projection based LoRA achieves similar with that training both $A$ and $B$, while needing less computation and memory cost.

**Rank of the LoRA.** A higher rank $r$ in LoRA means a greater number of trainable parameters and might lead to overfitting, while a lower rank $r$ means fewer trainable parameters and might lead to underfitting. When fine-tuning a pretrained model, if the dataset is significantly different and more complex, then it's would be better to use a high rank value (e.g., 64–256). On the other hand, if there doesn't involve a complex new dataset that the model hasn't encountered before, lower values of rank (e.g., 4–12) are sufficient. We conduct experiments under different settings of rank in LoRA (Severity-1). As shown in Table 13, our method is robust to the different ranks of LoRA, and we simply set $r = 4$ (which is a common choice for LoRA tuning) for all experiments in our main paper. In our case, we aim to teach the model learn reliability knowledge about the current task, without introducing a complex new dataset. Therefore, low value of rank like 4 or 2 is sufficient to learn the additional reliability knowledge without underfitting.

**TrustLoRA approximates a family of mappings.** One of the primary advantages of our method is to control the strength of each kind of reliability flexibly based on end-user preference. In Fig. 10, we

Table 14: Individual performance on the misclassification detection under covariate shifts, OOD detection under semantic shifts, and the unified unknown rejection on CIFAR-10/100 with severity-1.

| Method | ACC | MD-AURC | MD-FPR | MD-AUC | OOD-FPR | OOD-AUC | UR-AURC | UR-FPR | UR-AUC | F-AUC |
|---|---|---|---|---|---|---|---|---|---|---|
| | | | | CIFAR-10 | | | | | | |
| CE* | 87.86 | 35.33 | 42.33 | 88.61 | 39.96 | 86.24 | 56.04 | 37.53 | 89.49 | 87.41 |
| RegMixUp* | 89.27 | 34.01 | 55.70 | 87.65 | 50.18 | 86.08 | 57.05 | 50.60 | 88.56 | 86.86 |
| CRL* | 87.68 | 29.63 | 32.91 | 90.00 | 38.08 | 86.31 | 50.25 | 32.06 | 90.35 | 88.12 |
| LogitNorm* | 87.46 | 36.45 | 40.19 | 87.57 | 20.77 | 94.46 | 48.15 | 32.56 | 91.92 | 90.88 |
| OE* | 87.13 | 36.04 | 39.31 | 88.78 | 27.34 | 92.02 | 51.24 | 33.30 | 91.73 | 90.37 |
| RegMixUp (LoRA) | 88.48 | 36.62 | 55.64 | 87.52 | 54.89 | 84.74 | 62.17 | 53.63 | 87.97 | 86.11 |
| CRL (LoRA) | 88.14 | 35.04 | 43.20 | 88.40 | 41.03 | 86.36 | 56.28 | 38.40 | 89.36 | 87.37 |
| LogitNorm (LoRA) | 86.87 | 37.83 | 38.33 | 88.76 | 40.97 | 85.74 | 59.05 | 36.71 | 89.41 | 87.22 |
| AugMix (LoRA) | 90.53 | 18.05 | 33.51 | 90.47 | 41.92 | 87.88 | 36.62 | 36.09 | 90.72 | 89.16 |
| OE (LoRA) | 87.38 | 32.67 | 33.82 | 89.89 | 15.47 | 96.06 | 43.63 | 26.01 | 93.51 | 92.47 |
| Energy (AugMixLoRA) | 90.53 | 26.77 | 59.77 | 84.16 | 39.36 | 89.69 | 43.50 | 49.19 | 88.03 | 86.84 |
| TrustLoRA | 90.77 | 16.27 | 28.98 | 91.40 | 22.41 | 93.68 | **28.68** | **23.80** | **93.67** | **92.53** |
| | | | | CIFAR-100 | | | | | | |
| CE* | 66.08 | 134.63 | 49.00 | 84.74 | 70.45 | 69.44 | 163.66 | 54.88 | 82.95 | 76.33 |
| RegMixUp* | 68.64 | 115.84 | 48.72 | 85.10 | 69.27 | 69.84 | 154.87 | 54.34 | 83.12 | 76.72 |
| CRL* | 67.91 | 125.45 | 44.88 | 85.79 | 70.21 | 70.96 | 152.20 | 53.03 | 83.94 | 77.67 |
| LogitNorm* | 65.13 | 145.75 | 54.49 | 83.31 | 66.90 | 72.00 | 166.34 | 56.51 | 82.91 | 77.24 |
| OE* | 62.16 | 163.93 | 51.93 | 83.08 | 47.64 | 84.68 | 149.02 | 45.48 | 87.69 | 83.87 |
| RegMixUp (LoRA) | 67.53 | 126.07 | 50.51 | 84.54 | 66.98 | 72.39 | 155.58 | 53.51 | 83.81 | 77.99 |
| CRL (LoRA) | 67.44 | 115.43 | 48.90 | 84.85 | 66.20 | 73.16 | 153.48 | 52.10 | 84.27 | 78.57 |
| LogitNorm (LoRA) | 67.15 | 127.64 | 49.10 | 84.77 | 66.91 | 72.88 | 155.08 | 52.55 | 84.14 | 78.38 |
| AugMix (LoRA) | 70.48 | 102.55 | 49.80 | 84.52 | 63.66 | 75.39 | 141.23 | 51.24 | 84.69 | 79.69 |
| OE (LoRA) | 63.96 | 147.07 | 51.66 | 83.69 | 51.71 | 82.41 | 147.40 | 46.57 | 87.21 | 83.04 |
| Energy (AugMixLoRA) | 70.48 | 124.68 | 64.17 | 79.67 | 63.98 | 75.59 | 158.37 | 60.62 | 81.52 | 77.58 |
| TrustLoRA | 70.25 | 107.13 | 51.30 | 83.95 | 51.05 | 83.51 | **129.64** | **46.46** | **87.29** | **83.73** |

show that the scaling $\alpha$ in Eq. (7) can easily control the preference between MisD under covariate shits and OOD detection. Specifically, when increasing the $\alpha$, the OOD detection ability can be remarkably enhanced, and the Cov-MisD is increased at the beginning, and then decreased. Therefore, we can conclude that the proposed TrustLoRA can approximate a family of mappings with the simple linear combination of LoRA weights regarding different kinds of reliability knowledge flexibly.

# D  BROADER IMPACTS

Reliable prediction is an essential requirement for safe AI. The method proposed in this work would help the model detect unreliable prediction under both covariate and semantic shifts. Our work could contribute to the understanding of failure prediction in the wild. We are not aware of any negative social impact, and we believe the ethical aspects are not applicable.

**Limitations and Future Work.** Since our method is built on OE (Hendrycks et al., 2018), auxiliary outlier data is need. We have shown that TrustLoRA is robust to different auxiliary outlier data. Future work will consider developing outlier-free methods within the proposed framework. Besides, this paper only considers discriminative classifier in classification scenarios. Future work will explore unknown rejection in generative models, e.g., large language model and diffusion model, and more complex tasks such as Object detection and segmentation.

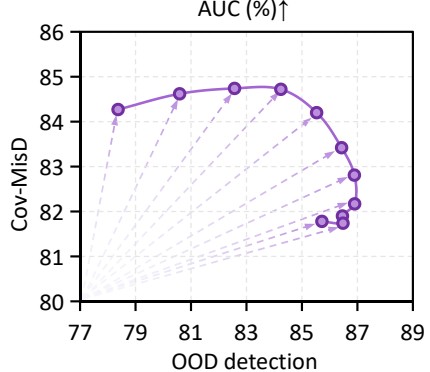

Figure 10: Cov-MisD and OOD detection performance with different addition weight.

