# OpenReview forum: "Towards Flexible and Controllable Unknown Rejection"
_ICLR.cc/2025/Conference — Submitted to ICLR 2025_

### Official Review · Reviewer_AfjX · 2024-10-20

**Soundness:** 2
**Presentation:** 3
**Contribution:** 2
**Rating:** 3
**Confidence:** 4

**Summary:**

This work introduces a novel reliability arithmetic framework using low-rank adaptation to separate and compress reliability knowledge, enabling unified and controllable rejection of unknowns under both covariate and semantic shifts, with experimental results showing its superiority over existing methods.

**Strengths:**

The presentation is fluent and the experimental results are sufficient.

**Weaknesses:**

1. From my perspective, the novelty of this paper seems somewhat limited, and the approach doesn't feel particularly compelling. For example, Equations (4) and (5) are essentially a combination of existing methods with LoRA, and this integration seems rather straightforward without much innovation.

2. Additionally, some parts of the paper might be a bit overly complicated in their presentation, though this is not a major issue. However, it could make the paper more challenging to understand for readers like myself. For instance, in the paragraph around line 180, the authors introduce the concept of “knowledge,” where they first acquire and then compress knowledge. However, there’s no clear formal definition of what this “knowledge” actually represents. In reality, it refers to the LoRA weights, and the so-called “compressed knowledge” is simply a linear combination of different LoRA weight increments. I feel that using the term “knowledge” here might make the understanding of the paper more difficult for readers.

3. The paper aims to address Unrecoverable, Inflexible, and Inefficient problems, but in my view, these issues are quite easily resolved by introducing LoRA and are not directly tied to the proposed concepts of knowledge acquisition and compression. This, in turn, limits the paper’s contribution in terms of innovation.

4. I’m having some difficulty understanding how the proposed method demonstrates superiority over others. Equations (4) and (5) are fairly common techniques, and the subsequent knowledge compression doesn't appear to significantly boost model performance. It seems to me that the main improvements might actually stem from the use of AugMix and auxiliary datasets rather than the method itself. For example, in most experiments, AugMix performs as the second-best method (only slightly behind the proposed one), while many other methods perform worse than this baseline. Therefore, it’s possible that the performance gains claimed by the paper could be largely due to AugMix, which is a strong baseline.

5. The paper devotes considerable space to discussing OOD generalization and detection, which, while important, doesn't seem to align closely with the core contribution of this work. I feel this aspect may not be directly related to the main method presented in the paper.

A few additional minor points:
1. There appears to be an issue with the citation or text on line 47: "Bai et al., (Bai et al., 2023)" may need to be corrected.
2. The numbering in the paper might be incorrect: (2) Recoverable should likely be (3) Recoverable.

**Questions:**

N/A

---

> ### Author Response · Authors · 2024-11-22
> **Reply to Reviewer AfjX (Part 1/3 )**
>
> We sincerely thank Reviewer AfjX for your review and are grateful for the time you spent on our submission. Below, we provide a point-by-point rebuttal to clarify your concerns.
>
> _**Q1. The novelty of this paper seems somewhat limited. The paper aims to address Unrecoverable, Inflexible, and Inefficient problems, but in my view, these issues are quite easily resolved by introducing LoRA.**_
>
> We agree with you that equations (4) (AugMix) and (5) (OE) are well-known, existing methods, which have been clearly described in the submission. However, as the Remark on Page 5, our contribution is not to design a specific method for OOD generalization or a specific method for OOD detection. Differently, we focus on the unified unknown rejection in the wild, where the key difficulty is how to get a model that can detect both misclassified covariate-shifted and semantic-shifted inputs in a unified and flexible manner because many recent methods fail to achieve this goal (e.g., many methods perform worse than the CE baseline).  In our paper, we demonstrate that _**combining existing OOD generalization and detection is not trivial**_, e.g., the straightforward multitask learning paradigm has limitations due to the objective conflict between the two objectives.  The proposed TrustLoRA is verified to be quite effective and flexible in the spirit of reliability separation and integration, outperforming multitask learning paradigm. We would like to clarify our contributions as follows.
> - **Problem setting.**  To the best of our knowledge, we might be _**the first to call for (1) flexible and controllable unknown rejection**_ that can control the uncertainty behavior of one's model, as deployment settings may evolve and change. Therefore, the topic of our paper is important, underexplored, interesting, and focused. **(2)** Rejecting both misclassified covariate-shifted and semantic-shifted samples in a unified manner,  are both very important for model trustworthiness.  There are only a few previous works that consider the recognition of covariate-shifts and rejecting semantic shifts, while they do not consider the rejection of misclassified covariate shifts.
> - **Problem analysis.** **(1)** In Section 4.1, we show the trade-off between the two unknown rejection tasks and verify that many popular existing methods are less effective for the unified unknown rejection problem. As shown in Fig.4, we can clearly observe that when fine-tuning with OE to acquire OOD detection ability, it is harder to detect misclassified corrupted samples. As a result, the unified unknown rejection performance also becomes worse. **(2)** On page 9, we analyze why multi-task learning is not the final solution for the unknown rejection problem. We believe _**those observations are insightful and enlightening.**_
> - **Proposed method.**  While the proposed TrustLoRA framework may appear natural and simple once explained, **(1)** to the best of our knowledge, we are _**the first to introduce LoRA into unknown rejection**_ or OOD detection fields.  Besides, the focus on parameter efficiency is also novel since efficiency is typically overlooked in the domain of uncertainty estimation, and TrustLoRA potentially presents a low-cost method of adding better uncertainty estimation ability to an existing model. The proposed random projection-based LoRA further improves the efficiency. **(2)** TurstLoRA is a general and simple framework. Although we implement it based on AugMix and OE, it can be built on other existing methods. Experimentally, our method _**outperforms strong baselines and the multi-task learning approach.**_
>
> **_The above contributions have been acknowledged by other Reviewers._** In summary, this work (1) studied an under-explored and important problem with desirable property for real-world application (**Reviewer Sug3, 8ZXm and 8Zzj**); (2) provides insights, offers important motivation for future research (**Reviewer 8Zzj**), and opens up exciting possibilities for future expansions (**Reviewer 8ZXm**); (3) the proposed TrustLoRA for unknown rejection is innovative and interesting (**Reviewer Sug3 and 8Zzj**), allowing flexible adaptation to deployment situations.

---

> > ### Author Response · Authors · 2024-11-22
> > **Reply to Reviewer AfjX (Part 2/3 )**
> >
> > _**Q2. Some parts of the paper might be a bit overly complicated in their presentation, though this is not a major issue.**_
> >
> > Thank you for your comment.  As demonstrated by Gueta et al., [A], knowledge can be represented by a region in weight space. In line 180, we introduce the concept of “reliability knowledge”, which refers to the ability of unknown rejection of failure cases like misclassified covariate shifts and semantic shifts.
> > - We intend to distinguish it from the ordinary ability of classification or recognition.  After fine-tuning the model with a failure-specific learning objective, the points in weight space embed the necessary knowledge to perform the task and can be viewed as the corresponding knowledge. Following your valuable comment, we modify some descriptions on Page 4.
> > - As shown in the experiments (e.g., Fig.5 and Fig.6), a linear combination of different LoRA weights in our framework can achieve the purpose of addition and negation of specific unknown rejection ability, without affecting the ID classification accuracy. Therefore, we can conclude that the LoRA weight in our method encodes the corresponding reliability knowledge, enabling flexible and controllable reliability edition.
> >
> > **Reference**
> >
> > _[A] Gueta, Almog, et al. Knowledge is a Region in Weight Space for Fine-tuned Language Models, EMNLP 2023._

---

> > > ### Author Response · Authors · 2024-11-22
> > > **Reply to Reviewer AfjX (Part 3/3 )**
> > >
> > > _**Q3. How the proposed method demonstrates superiority over others. It’s possible that the performance gains claimed by the paper could be largely due to AugMix.**_
> > >
> > > **(1)** The most important metric for unknown rejection is AURC. As shown in Table 1&2, TrustLoRA remarkably outperforms AugMix and OE, and other recent approaches such as OpenMix, SURE, RCL and SCONE, and we list those results below.
> > >
> > > **(2)** Particularly, as shown in Table 5, our method performs better than multi-task learning (which is unrecoverable, inflexible, and inefficient) of AugMix and OE, and also outperforms SIRC (AugMix-based). Those results demonstrate that how to combine OOD generalization (e.g., AugMix) and detection (e.g., OE) methods are crucial for the challenging unified unknown rejection problem studied in our paper.
> > >
> > > **(3)**  The fact that many other methods perform worse than AugMix further demonstrates the challenge of our proposed unified unknown rejection problem, and we believe that our work made a step towards this problem.
> > > | CIFAR-10        | AURC (S1) | FPR95 (S1) | AUC (S1) | F-AUC (S1) | AURC (S2) | FPR95 (S2) | AUC (S2) | F-AUC (S2) | AURC (S3) | FPR95 (S3) | AUC (S3) | F-AUC (S3) |
> > > |--------------------|------------|-------------|-----------|-------------|------------|-------------|-----------|-------------|------------|-------------|-----------|-------------|
> > > | OpenMix$^{\ast}$  | 29.46      | 28.13       | 92.45     | 91.08       | 46.90      | 32.61       | 90.95     | 89.12       | 66.86      | 36.94       | 89.18     | 86.83       |
> > > | SURE$^{\ast}$     | 31.25      | 27.39       | 92.67     | 91.26       | 48.13      | 31.19       | 91.02     | 89.51       | 68.31      | 36.60       | 89.55     | 86.27       |
> > > | RCL$^{+}$         | 58.01      | 35.92       | 89.53     | 87.47       | 93.19      | 43.11       | 86.95     | 84.17       | 132.03     | 48.36       | 84.57     | 81.12       |
> > > | SCONE$^{+}$       | 44.01      | 26.99       | 93.13     | 92.08       | 71.28      | 32.75       | 91.17     | 89.37       | 104.06     | 38.91       | 88.83     | 86.15       |
> > > |AugMix |36.62 | 36.09 | 90.72  | 89.16 |51.73 | 39.06 | 89.61  | 87.71 | 67.82  | 41.97 | 88.51  | 86.20 |\\
> > > | FS-KNN (AugMix)            | 47.54      | 55.75       | 87.29     | 86.01       | 62.36      | 55.35       | 86.79     | 85.30       | 80.36      | 57.86       | 85.52   | 83.74       |
> > > | **TrustLoRA**     | **28.68**  | **23.80**   | **93.67** | **92.53**   | **41.64**  | **27.43**   | **92.67** | **91.18**   | **56.64**  | **30.62**   | **91.55** | **89.65**   |
> > >
> > > | CIFAR-100               | AURC (S1) | FPR95 (S1) | AUC (S1) | F-AUC (S1) | AURC (S2) | FPR95 (S2) | AUC (S2) | F-AUC (S2) | AURC (S3) | FPR95 (S3) | AUC (S3) | F-AUC (S3) |
> > > |-----------------------|----------|-----------|---------|-----------|----------|-----------|---------|-----------|----------|-----------|---------|-----------|
> > > | OpenMix$^{\ast}$ | 134.18   | 46.46     | 86.57   | 81.84     | 164.32   | 51.49     | 84.47   | 78.78     | 203.87   | 56.18     | 82.29   | 76.10     |
> > > | SURE$^{\ast}$    | 137.62   | 48.35     | 86.04   | 82.26     | 172.45   | 51.76     | 83.95   | 78.41     | 205.71   | 56.62     | 82.18   | 76.54     |
> > > | RCL$^{+}$        | 155.85   | 52.89     | 84.29   | 78.73     | 202.39   | 59.20     | 81.61   | 74.88     | 256.47   | 64.10     | 79.11   | 71.83     |
> > > | SCONE$^{+}$     | 148.50   | 47.65     | 86.50   | 81.69     | 201.73   | 53.92     | 83.57   | 77.37     | 264.59   | 59.29     | 80.74   | 74.03     |
> > > |AugMix |141.23 | 51.24 | 84.69 | 79.69 | 158.07 | 54.44 | 83.47 | 77.63 | 177.71 | 57.12 | 82.26 | 75.69 |
> > > | FS-KNN (AugMix) | 143.83 | 56.08 | 84.71 | 81.69 | 167.36 | 58.13 | 83.03 | 79.74 | 183.17 | 59.21 | 82.26 | 77.12 |
> > > | **TrustLoRA**        | **129.64** | **46.46** | **87.29** | **83.73** | **149.14** | **50.12** | **85.77** | **81.40** | **172.35** | **53.32** | **84.41** | **79.28** |
> > >
> > >
> > > _**Q4. Discussing OOD generalization and detection may not be directly related to the main method presented in the paper.**_
> > >
> > > Thank you for your comment. The problem we studied, unknown rejection, is related to OOD generalization and detection, but with notable differences. Compared with them, the unknown rejection formulated in our paper is more practical and challenging, offering important motivation for future research. Following your suggestion, we removed the related work section to the Appendix and added more recent references suggested by Reviewer 8Zzj.
> > >
> > > _**Q5. There appears to be an issue with the citation or text on line 47: "Bai et al., (Bai et al., 2023)" may need to be corrected. The numbering in the paper might be incorrect: (2) Recoverable should likely be (3) Recoverable.**_
> > >
> > > Thank you for your careful reading, and we corrected the typo in the revised paper.

---

> > > > ### Comment · Reviewer_AfjX · 2024-11-23
> > > >
> > > > I think the proposed method can be roughly summarized with a Equation: $$LoRA + AugMix + OE.$$  I will not change my score.

---

> ### Author Response · Authors · 2024-11-23
>
> Dear Reviewer AfjX,
>
> Thanks for your feedback. In our previous response (Part 1/3), we summarize the contribution of this paper in three aspects, i.e., problem setting, problem analysis, and proposed method. Your major concern is about the proposed method.
>
> AugMix and OE are failure-specific methods.  In recent years, various failure-specific, especially OOD detection methods, have been proposed. However, they are useless or even harmful for real-world unknown rejection when facing both covariate and semantic shifts.  We respectfully argue that investigating how to better use existing failure-specific techniques is meaningful and valuable, perhaps more important than proposing another failure-specific approach.
>
> This work aims to design a unified framework to integrate different sources of reliability knowledge. We would like to emphasize that ensembling different failure-specific objectives is not trivial. For example,  optimizing AugMix and OE in the multi-task learning manner is less effective (e.g., AugMix + OE is worse than OE) due to the existing conflicts (as shown in the Table below, CIFAR-100 with S1), and also suffers from unrecoverable, inflexible, and inefficient issues. In this work, we modify the original LoRA with random projection (which is more computation and memory efficient) and tune it with different failure-specific objectives sequentially. Finally, flexible and controllable unknown rejection can be achieved with LoRA arithmetic.
> Therefore, this work is not simple LoRA + AugMix + OE.  We believe our work shows the good application of LoRA for the important and challenging unified unknown rejection problem.
>
> |Method|AURC ($\downarrow$) | FPR95 ($\downarrow$) | AUC ($\uparrow$) | F-AUC ($\uparrow$) |
> |------|------|---------|--------|--------|
> |OE | 149.02|**45.48**  | **87.09** |**83.17**  |
> |AugMix|**133.86**|49.79  |85.05|80.48 |
> |AugMix+OE|138.92  |50.26  |85.40  | 81.44|
>
> We hope the above responses and our previous responses have addressed your concerns. As other Reviewers mentioned, this work focuses on an under-explored and important problem (Sug3, 8ZXm, 8Zzj), provides valuable insights, offers important motivation for future research (8Zzj, 8ZXm), and the introduction of LoRA into unknown rejection field is innovative and interesting (Sug3, 8Zzj, 8ZXm). If you still have some specific concerns, please let us know. We look forward to and appreciate your feedback.

---

> > ### Comment · Reviewer_AfjX · 2024-11-23
> >
> > Thank you for your response. I would like to know if you agree with what I mentioned. From a purely methodological perspective, the method presented in the paper is mainly $LoRA + OE + AugMix$.

---

> > > ### Author Response · Authors · 2024-11-23
> > >
> > > Dear Reviewer AfjX
> > >
> > > Thanks for your feedback. We respectfully disagree that our method is simple LoRA + AugMix + OE. If I understand you correctly, the AugMix + OE in your summarized equation means optimizing the learning objectives of AugMix and OE in a multi-task learning manner, which is the most straightforward way. While we show the multi-task learning manner is less effective, and thus propose to learn different failure-specific objectives sequentially and consolidate them via LoRA addition or negation. Besides, since we call for flexible, efficient, and recoverable, we further modify the original LoRA with random projection, which is more computation and memory efficient. Therefore, we respectfully argue that our method is not simple LoRA + AugMix + OE.

---

> > > > ### Comment · Reviewer_AfjX · 2024-11-23
> > > >
> > > > So could you summarize your methodological contribution in one sentence (without mentioning Lora)?

---

> > > > > ### Author Response · Authors · 2024-11-23
> > > > >
> > > > > Dear Reviewer AfjX
> > > > >
> > > > > Thanks for your feedback. Our methodological contribution can be summarized as follows: _Towards flexible and controllable unknown rejection, we propose a two-stage reliability knowledge separation and integration framework, which overcomes the limitation of failure-specific full training and multi-task learning._ In practice, the reliability knowledge separation stage is realized by LoRA-based failure-specific tuning, and the second stage is realized by LoRA arithmetic.

---

> ### Comment · Reviewer_AfjX · 2024-11-23
>
> Thank you for your response, and I respect your opinion, but I still disagree with your viewpoint. The contribution you mentioned is more of a framework-level contribution rather than a methodological one. I still believe that merely proposing such a so-called framework might be insufficient in terms of methodology. For instance, the methodological contribution of LoRA, which you cited, can be clearly articulated in terms of methodology .

---

### Official Review · Reviewer_8Zzj · 2024-11-02

**Soundness:** 2
**Presentation:** 3
**Contribution:** 2
**Rating:** 5
**Confidence:** 5

**Summary:**

This study proposes a learning approach that integrates OOD generalization and OOD detection to enable more reliable deployment of deep neural networks in real-world applications. To effectively distinguish correctly classified samples from incorrectly classified covariate-shifted and semantically shifted samples, the authors introduce a strategy that merges LoRA weights trained on each failure case. The proposed method demonstrates superior unknown rejection performance, validated on CIFAR-10/100 and ImageNet-200/500 benchmarks.

**Strengths:**

- This study addresses the unknown rejection task, a desirable property required in real-world applications where inputs come from various sources at test time, and proposes an effective solution. This approach offers a useful methodology for practitioners.
- The use of LoRA for unknown rejection is interesting. LoRA is parameter-efficient and portable, allowing flexible adaptation to deployment situations. Additionally, the merging of LoRA weights with different objectives demonstrates the effective handling of multi-objective scenarios.
- Through extensive experiments, the study provides insights into the unknown rejection task. For example, the impact of covariate shifts on MisD and OOD detection (Fig. 2) and the trade-off between rejecting misclassified covariate-shifted samples versus semantic-shifted samples (Fig. 4) highlight the challenges of the unknown rejection task and offer important motivation for future research.

**Weaknesses:**

- The primary concern with this study is the insufficient review of prior research on unknown rejection. Even though previous works may not have integrated OOD generalization and OOD detection as stated in the motivation of this study, there are several existing studies focused on developing robust models from an unknown rejection perspective. To rigorously assess the advantages of the proposed method, discussions and comparative experiments with prior works on unknown rejection or failure detection are necessary:
  - Zhu et al. “RCL: Reliable continual learning for unified failure detection”, CVPR 2024
  - Cen et al. “The devil is in the wrongly-classified samples: Towards unified open-set recognition”, ICLR 2023
  - Zhu et al. “OpenMix: Exploring outlier samples for misclassification detection”, CVPR 2023
  - Li et al. “SURE: Survey recipes for building reliable and robust deep networks”, CVPR 2024
- The failure rejection setting in this study does not align with requirements for real-world environments. In this paper, the main experiments in Tables 1 and 2 aim to distinguish between “correctly classified covariate-shifted” and “misclassified covariate-shifted + semantic shift” cases. However, in a real inference environment, ID, covariate-shifted, and semantic-shifted samples may all be input to the model. For the model to be reliable in such settings, it should effectively distinguish between “correctly classified ID + correctly classified covariate-shifted” and “misclassified ID + misclassified covariate-shifted + semantic shift” samples. Although the performance on clean ID data is discussed on page 8, including ID test data in the main experimental setup would better align the study with realistic scenarios.

**Questions:**

- In Fig. 1, it is necessary to specify what each red cross and green check mark represents.
- On page 5, it is mentioned that only the B matrix of LoRA is trained, but since LoRA itself does not have many parameters, it is unclear why efficiency is so crucial. Are there other advantages besides efficiency? Does training both A and B affect performance?
- In the CIFAR-10/100 experiments, it is not specified which data was used as $D_{aux}$.
- Proposition 3.1 suggests that TrustLoRA can approximate a family of mappings $f_n$, but can this be guaranteed solely through the linear combination of LoRA weights?
- Typos
  - A closing bracket is missing in Proposition 3.1.
  - Line 318: “generally keep equal numbers of misclassified semantic-shifted data” → should be “generally keep equal numbers of misclassified covariate-shifted data.”
  - Line 485: The reference “Narasimhan et al.” is missing the publication year.
  - Line 491: “Fig. 8” should be “Fig. 7.”

---

> ### Author Response · Authors · 2024-11-22
> **Reply to Reviewer 8Zzj (Part 1/2 )**
>
> We sincerely thank Reviewer 8Zzj  for your review and are grateful for the time you spent on our submission. We are glad for the acknowledgment that our work provides insights into the unknown rejection task and the proposed approach is interesting, effective and practical to tackling a critical challenge in real-world applications. Below we would like to give detailed responses to each of your comments.
>
> _**Q1. Insufficient review of prior research on unknown rejection.. discussions and comparative experiments with prior works [A, B, C, D] are necessary.**_
>
> Thank you for this comment and suggested references. Following your suggestion, we added discussions and comparisons with those prior works (_OpenMix [C], SURE [D], RCL [A], FS-KNN [B]_) on unknown rejections in the revised version.
> - In the  Introduction section on Page 2, we discuss our work with them and add a new paragraph in the Related work section in the Appendix, providing detailed discussion.
> - Experimental comparison have been conducted and added to Tables 1&2 on Pages 7 and 8 of the revised version, and we summarize the results below. It is observed that TrustLoRA can outperform those strong baselines.
>
> | CIFAR-10        | AURC (S1) | FPR95 (S1) | AUC (S1) | F-AUC (S1) | AURC (S2) | FPR95 (S2) | AUC (S2) | F-AUC (S2) | AURC (S3) | FPR95 (S3) | AUC (S3) | F-AUC (S3) |
> |--------------------|------------|-------------|-----------|-------------|------------|-------------|-----------|-------------|------------|-------------|-----------|-------------|
> | OpenMix$^{\ast}$  | 29.46      | 28.13       | 92.45     | 91.08       | 46.90      | 32.61       | 90.95     | 89.12       | 66.86      | 36.94       | 89.18     | 86.83       |
> | SURE$^{\ast}$     | 31.25      | 27.39       | 92.67     | 91.26       | 48.13      | 31.19       | 91.02     | 89.51       | 68.31      | 36.60       | 89.55     | 86.27       |
> | RCL$^{+}$         | 58.01      | 35.92       | 89.53     | 87.47       | 93.19      | 43.11       | 86.95     | 84.17       | 132.03     | 48.36       | 84.57     | 81.12       |
> | SCONE$^{+}$       | 44.01      | 26.99       | 93.13     | 92.08       | 71.28      | 32.75       | 91.17     | 89.37       | 104.06     | 38.91       | 88.83     | 86.15       |
> | FS-KNN (AugMix)           | 47.54      | 55.75       | 87.29     | 86.01       | 62.36      | 55.35       | 86.79     | 85.30       | 80.36      | 57.86       | 85.52   | 83.74       |
> | **TrustLoRA**     | **28.68**  | **23.80**   | **93.67** | **92.53**   | **41.64**  | **27.43**   | **92.67** | **91.18**   | **56.64**  | **30.62**   | **91.55** | **89.65**   |
>
> | CIFAR-100               | AURC (S1) | FPR95 (S1) | AUC (S1) | F-AUC (S1) | AURC (S2) | FPR95 (S2) | AUC (S2) | F-AUC (S2) | AURC (S3) | FPR95 (S3) | AUC (S3) | F-AUC (S3) |
> |-----------------------|----------|-----------|---------|-----------|----------|-----------|---------|-----------|----------|-----------|---------|-----------|
> | OpenMix$^{\ast}$ | 134.18   | 46.46     | 86.57   | 81.84     | 164.32   | 51.49     | 84.47   | 78.78     | 203.87   | 56.18     | 82.29   | 76.10     |
> | SURE$^{\ast}$    | 137.62   | 48.35     | 86.04   | 82.26     | 172.45   | 51.76     | 83.95   | 78.41     | 205.71   | 56.62     | 82.18   | 76.54     |
> | RCL$^{+}$        | 155.85   | 52.89     | 84.29   | 78.73     | 202.39   | 59.20     | 81.61   | 74.88     | 256.47   | 64.10     | 79.11   | 71.83     |
> | SCONE$^{+}$     | 148.50   | 47.65     | 86.50   | 81.69     | 201.73   | 53.92     | 83.57   | 77.37     | 264.59   | 59.29     | 80.74   | 74.03     |
> | FS-KNN (AugMix)            | 47.54      | 55.75       | 87.29     | 86.01       | 62.36      | 55.35       | 86.79     | 85.30       | 80.36      | 57.86       | 85.52   | 83.74       |
> | **TrustLoRA**        | **129.64** | **46.46** | **87.29** | **83.73** | **149.14** | **50.12** | **85.77** | **81.40** | **172.35** | **53.32** | **84.41** | **79.28** |
>
>
> **Reference**
>
> _[A] Zhu et al. “RCL: Reliable continual learning for unified failure detection”, CVPR 2024._
>
> _[B] Cen et al. “The devil is in the wrongly-classified samples: Towards unified open-set recognition”, ICLR 2023._
>
> _[C] Zhu et al. “OpenMix: Exploring outlier samples for misclassification detection”, CVPR 2023._
>
> _[D] Li et al. “SURE: Survey recipes for building reliable and robust deep networks”, CVPR 2024._

---

> ### Author Response · Authors · 2024-11-22
> **Reply to Reviewer 8Zzj (Part 2/2 )**
>
> _**Q2. The failure rejection setting in this study does not align with requirements for real-world environments... including ID test data would better align the study with realistic scenarios.**_
>
> Thank you for your valuable suggestion. In the initial submission, we did not include clean ID in Table 1&2 in order to clearly reflect the unknown rejection ability under covariate and semantic shits. Following your suggestion:
> - We added a clear description in Datasets and implementation part on Page 6.
> - To comprehensively present and discuss the results when clean ID is involved, we **(1)** include a new Figure 7 about the MisD performance on clean ID and **(2)** report the unified unknown rejection performance evaluated follow your instruction in Table 7 on Page 10 in the revised version. We also report the results below (Table 7) and find that our method performs best. We believe that the newly added results and discussion would better align the study with realistic scenarios.
>
> | Method        | AURC (CIFAR-10) | FPR95 | AUC | F-AUC| AURC (CIFAR-100) | FPR95 | AUC | F-AUC|
> |---------------|-----------------|------------------|----------------|------------------|------------------|------------------|-----------------|-------------------|
> | CE            | 32.62           | 31.56            | 91.59          | 89.98            | 134.19           | 48.70            | 85.47           | 79.71             |
> | RegMixUp      | 34.82           | 44.92            | 90.56          | 89.27            | 133.88           | 49.25            | 84.96           | 79.35             |
> | CRL           | 30.37           | 26.24            | 92.30          | 90.54            | 129.75           | 47.73            | 85.97           | 80.47             |
> | LogitNorm     | 27.11           | 26.10            | 93.76          | 92.97            | 138.73           | 51.38            | 85.22           | 80.20             |
> | AugMix        | 27.47           | 31.75            | 91.85          | 90.45            | 133.73           | 49.50            | 85.34           | 80.70             |
> | MaxLogit      | 32.53           | 46.17            | 89.95          | 88.86            | 142.31           | 56.11            | 84.00           | 79.74             |
> | Energy        | 35.48           | 50.48            | 88.52          | 87.49            | 149.95           | 58.96            | 82.23           | 78.55             |
> | GEN           | 33.33           | 46.73            | 89.31          | 88.31            | 149.52           | 58.66            | 82.30           | 78.59             |
> | ASH           | 33.09           | 46.95            | 89.67          | 88.61            | 142.67           | 56.05            | 83.87           | 79.75             |
> | **TrustLoRA** | **20.86**       | **20.86**        | **94.42**      | **93.47**        | **121.90**       | **44.44**        | **87.75**       | **84.35**         |
>
>
> _**Q3. Specify what each red cross and green check mark represents.**_
>
> Thank you for your comment. We have added the meaning of the red cross and green check in the caption of Figure 1.
>
> _**Q4.  Are there other advantages besides efficiency of training B matrix of LoRA? Does training both A and B affect performance?**_
>
> For LoRA, tuning only B matrix mainly aim to further improve the efficiency of LoRA without affecting the performance. In below, we compare the original LoRA and our random projection based LoRA, and find that they perform similar. We have added those results in Table 12 in the Appendix.
> | (CIFAR-10)| AURC | FPR95 | AUC | F-AUC |(CIFAR-100)| AURC | FPR95 | AUC | F-AUC |
> |------|---------|--------|--------|--------|------|---------|--------|--------|--------|
> | B only | 28.68 | 23.80 | 93.67 | 92.53 |B only | 129.64 | 46.46 | 87.29 | 83.73 |
> | A \& B | 28.07 | 24.12 | 93.82 | 92.95|A \& B | 127.51 | 45.72 | 87.17 | 83.34 |
>
>
> _**Q5. Linear combination of LoRA weights for approximating a family of mappings.**_
>
> We provide Figure 9 in the Appendix of the revised paper to show that TrustLoRA can approximate a family of mappings, where scaling $\alpha$ can control the preference between MisD under covariate shits and OOD detection. Figure 6 shows that we can achieve selective reliability forgetting with LoRA negation. Those results imply the effectiveness of the simple linear combination of LoRA weights once we compress the reliability knowledge in LoRA module.
>
> _**Q6. Typos.**_
>
> Thank you for your careful reading, and we corrected the typo in the revised paper.

---

> > ### Author Response · Authors · 2024-11-25
> > **Further comments and discussions will be appreciated**
> >
> > Dear Reviewer 8Zzj,
> >
> > Thank you for your valuable time to review our work and for your constructive feedback. We posted our response to your comments a few days ago with our latest experimental results, and we wonder if you could kindly share some of your thoughts so we can address your concern if there are any.
> >
> > In the previous response,
> > 1.  We have added discussions and preliminary experiments with those prior works (OpenMix, SURE, RCL, FS-KNN) on unknown rejections in the revised version, which shows that TrustLoRA can outperform those strong baselines.
> > 2. Following your suggestion, we report the unified unknown rejection performance evaluated following your instruction in Table 7 on Page 10 in the revised version. We believe the newly added results and discussion would better align the study with realistic scenarios.
> > 3. We added experiments to show that training the B matrix of LoRA is the same effective as training both A and B.
> >
> > We would appreciate it if you could kindly take a look at both the revision and our response to your comments. If you have any further questions, we are happy to discuss them!
> >
> > Best regards,
> >
> > Authors

---

> > > ### Comment · Reviewer_8Zzj · 2024-11-25
> > >
> > > I sincerely appreciate the authors' detailed responses. Most of my concerns have been addressed. The comparison with prior works on unknown rejection is well-covered, and the ablation study on LoRA training is a helpful addition.
> > >
> > > However, I still have a couple of remaining concerns:
> > > - While I appreciate the empirical demonstration showing how the linear combination of LoRA weights approximates a family of mapping, my main question was about the rigor of Proposition 3.1. The authors reference Dimitriadis et al. (2023) regarding Pareto optimality, but the proof in Appendix B.1 does not clearly explain whether replacing $\theta_1$ and $\theta_2$ with LoRA weights satisfies the proposition. For instance, Zeng and Lee (2024) outline several conditions for LoRA to be a universal approximator. It would be helpful to clarify the exact conditions under which Proposition 3.1 holds.
> > > -  While additional experiments involving ID data in realistic scenarios have been included, I still believe that the scenario (“correctly classified ID + correctly classified covariate-shifted” vs. “misclassified ID + misclassified covariate-shifted + semantic shift”) should be the primary focus of the paper. It is disappointing that more extensive experiments in this scenario were not conducted. I understand that addressing this fully during the rebuttal period may be challenging, but I want to emphasize the mismatch between the settings the paper aims to address and the main experiments presented.
> > >
> > > For these reasons, I will maintain my current rating.
> > >
> > > *Zeng and Lee (2024), "The expressive power of low-rank adaptation", ICLR*

---

> > > > ### Author Response · Authors · 2024-11-26
> > > > **Reply to Reviewer 8Zzj (Remaining Concern, Part 1/2 )**
> > > >
> > > > Dear Reviewer 8Zzj,
> > > >
> > > > Thank you for your valuable feedback. We would like to begin by addressing your second question. The experimental results in Table 7 on Page 10 indeed represent the  results of “correctly classified ID + correctly classified covariate-shifted”  vs. “misclassified ID + misclassified covariate-shifted + semantic shift,” as you suggested.
> > > >
> > > > We understand that you may prefer us to replace all the experiments in our submission with this setting. However, our intention with the current experimental setup in Tables 1 and 2 is to specifically highlight the unknown rejection ability under covariate and semantic shifts, as we have been challenged by others that mixing all scenarios together might obscure the effectiveness of the method for individual failure cases. Therefore, we have chosen to focus on covariate and semantic shifts in Tables 1 and 2 (with individual performance also presented in Table 13 of the appendix), while providing additional results for the scenario “correctly classified ID + correctly classified covariate-shifted” vs. “misclassified ID + misclassified covariate-shifted + semantic shift.” Additional, we also report results for clean ID scenarios as a reference. We believe that these newly added experiments clearly demonstrate the effectiveness of our method in handling clean ID data, covariate shifts, and semantic shifts.
> > > >
> > > > Additionally, the rejection of misclassified IDs, along with the trade-off between ID misclassification detection and out-of-distribution (OOD) detection, has been extensively studied in recent works (e.g., Jaeger et al., 2022; Narasimhan et al., 2024; Zhu et al., 2023a; Cen et al., 2023). Our work, which focuses on unknown rejection under covariate and semantic shifts, complements these existing studies.
> > > >
> > > > We kindly ask for your understanding regarding the challenges of replacing all experiments and rewriting the entire experimental section. Apart from the presentation concerns, if you have any questions related to the unknown rejection performance of our method on ID data, covariate shifts, or semantic shifts, please feel free to let us know, and we would be happy to discuss them.
> > > >
> > > > Best regards,
> > > > The Authors

---

> > > > > ### Author Response · Authors · 2024-11-26
> > > > > **Reply to Reviewer 8Zzj (Remaining Concern, Part 2/2 )**
> > > > >
> > > > > Dear Reviewer 8Zzj,
> > > > >
> > > > > For the first remaining concern about the rigor of Proposition 3.1:
> > > > >
> > > > > (1) In Appendix B1, $\theta_i$ ($i=1,2$)  represents the difference between the fine-tuned model and the pre-trained model, rather than a different new model. Therefore, the low-rank module is used to approximate the residual parameters, i.e., $\theta^l_i-\theta^l_{pre} = B^l_i A^l_i$, where the $i$ represents the layer index. In our work, we fine-tune a pre-trained model (with good classification ability) to acquire reliability knowledge for specific failure cases like covariate and semantic shifts. Thus, the discrepancy between the pre-trained model and the fine-tuned model $(\theta^l_i-\theta^l_{pre} = B^l_i A^l_i)^{L}_{l=1}$ is relatively small.
> > > > >
> > > > > To quantify the above point, we calculated the cos similarity of the layer-wise parameter between the pre-trained model (ResNet-18, CIFAR-100) and the fine-tuned model with OE, and the similarities are generally $>0.95$, and this is also true for AugMix fine-tuned model. We added a new Figure 8 in the Appendix to show the layer-wise similarity and also illustrate the classification region, covariate, and semantic reliability region. Therefore, we could say that the LoRA module has the expressive power to approximate the discrepancy between those regions.
> > > > >
> > > > > (2) In Zeng and Lee (2024), the authors outline several conditions for LoRA to be an **exact** universal approximator. When the employed LoRA rank is lower than the critical threshold, the authors provide an upper bound for the approximation error in Theorem 5. Specifically, the approximation error is related to i) the magnitude of the target model’s parameters and the input; ii) the rank of the adapter and the discrepancy between the frozen model and the target model; iii) the depth of the frozen model $L$. Besides, as shown in the experiments of Zeng and Lee (2024), LoRA with $rank=4$ already has quite a small approximation error.
> > > > > In our work, our proven in Appendix B1 is based on the universal approximation theorem (Haykin, 1998) with unconstrained width of the LoRA module (e.g, D and Q), and the $ϵ$ expresses the approximation error (we do not focus on exactly universal approximator). Inspired by your valuable comment, we added a remark about this point after the proof in Appendix B1.
> > > > >
> > > > > If you have any other questions, please feel free to let us know, and we would be happy to discuss them.
> > > > >
> > > > > Best regards,
> > > > > The Authors

---

> > > > > > ### Comment · Reviewer_8Zzj · 2024-11-28
> > > > > >
> > > > > > I sincerely appreciate the authors’ detailed responses. Below are my comments regarding your reply:
> > > > > >
> > > > > > **For Remaining Concern, Part 1/2:**
> > > > > >
> > > > > > First, I understand that revising the main experiments or significantly modifying the manuscript within the limited rebuttal period is not feasible. As a reviewer, I am not requesting substantial changes at this stage. However, I would like to highlight the inconsistency between the problems the manuscript aims to address (as well as the scope of problems addressed in prior works) and the main experiments conducted. In Figure 1, unknown rejection is defined as *“rejecting misclassified covariate-shifted and all semantic-shifted OOD samples”*. However, prior studies, such as Jaeger et al. (2022), Kim et al. (2023b), and Narasimhan et al. (2024), have included misclassified ID samples in their setups, naming these tasks failure detection or unknown detection. It is unfortunate that this work does not expand upon the directions explored in previous research.
> > > > > >
> > > > > > Additionally, in Section 3.1 (Motivation), the paper states:
> > > > > > *“It is acknowledged that rejecting incorrect predictions is essential for reliable learning. However, the failure sources are rich in uncontrolled environments, including incorrect predictions of ID or corrupt-shifted samples, and also inputs from unknown new categories”*.
> > > > > > I fully agree with this statement, which is why I believe that the main experiments not being centered on this scenario reflect a logical inconsistency in the manuscript’s structure. While additional experiments for this scenario were included, they were not the primary focus, which I find unfortunate. In the response, the authors noted that “mixing all scenarios together might obscure the effectiveness of the method for individual failure cases”, but this could be addressed by performing additional analyses for each failure type.
> > > > > >
> > > > > > **For Remaining Concern, Part 2/2:**
> > > > > >
> > > > > > Thank you for providing additional empirical validation for Proposition 3.1 and discussing the findings from Zeng and Lee (2024). However, I still have the following questions and would appreciate clarification if I have misunderstood.
> > > > > > - Does Proposition 3.1 imply that $\theta_{pre}$ is fixed and that any $f$ can be approximated through the linear combination of LoRA weights? From the proof, I understand this is not the case. Zeng and Lee (2024) analyze the approximation power of LoRA under the assumption that $\theta_{pre}$ is a frozen parameter.
> > > > > > - If $\theta_{pre}$ is not fixed, how does Proposition 3.1 theoretically support the proposed method? Without the linear combination term in LoRA, it seems to reduce to the universal approximation theorem. If this is the case, what additional value does the linear combination term in LoRA provide beyond the universal approximation theorem?

---

### Official Review · Reviewer_8ZXm · 2024-11-03

**Soundness:** 3
**Presentation:** 3
**Contribution:** 4
**Rating:** 8
**Confidence:** 3

**Summary:**

This paper addresses the significant challenge of deploying deep neural networks in open environments where models may encounter both **covariate shifts**—changes in the input distribution while the label space remains the same—and **semantic shifts**, where inputs belong to new, unseen classes. Traditionally, methods have dealt with Out-of-Distribution (OOD) generalization (handling covariate shifts) and OOD detection (identifying semantic shifts) separately. However, in practical applications, a model needs to accept correctly classified inputs while rejecting both misclassified covariate-shifted examples and all semantic-shifted samples.

To solve this problem, the authors propose a framework that leverages **Low-Rank Adapters (LoRA)** to acquire and compress failure-specific reliability knowledge. By fine-tuning LoRA modules on specific tasks—such as handling covariate shifts using AugMix and semantic shifts using Outlier Exposure (OE)—the model can flexibly combine these modules during inference through simple arithmetic operations like addition and scaling. This approach allows for a controllable and efficient method to enhance unknown rejection capabilities under both types of distributional shifts.

**Problem Definition and Proposed Method:**

As summarized above, the paper addresses the challenge of unknown rejection in open environments, where a model must handle both covariate shifts, which are changes in the input distribution, and semantic shifts, involving new, unseen classes. During the training phase, the model learns from in-distribution data composed of pairs of inputs and their corresponding labels. This data is sampled from a specific distribution over a defined input and label space, including multiple classes. The classifier operates by selecting the class with the highest confidence score produced by the model for a given input, effectively determining the input's class based on the model's assessment of each possible class.

In deployment, the model encounters data from covariate shifts $ P_{\text{covariate}}^{\text{out}} $ (with shifted input distribution but same labels) and semantic shifts $ P_{\text{semantic}}^{\text{out}} $ (inputs from unknown classes $ y \notin Y $). The goal is to accept correctly classified samples from both $ P_{\text{in}} $ and $ P_{\text{covariate}}^{\text{out}}$, and to reject misclassified samples and all samples from $ P_{\text{semantic}}^{\text{out}} $. To achieve this, the authors propose using Low-Rank Adapters (LoRA) to capture failure-specific knowledge. LoRA adds low-rank matrices $ A $ and $ B $ to the model's layers, modifying the forward pass to $ z = W x + B A x $, where $ W $ are the original weights (kept frozen) and $ r = \text{rank}(A) \ll \min(u, v) $. Separate LoRA modules are fine-tuned for covariate shifts (using AugMix) and semantic shifts (using Outlier Exposure with auxiliary data $ D_{\text{aux}} $). These modules are then combined using $ \tau = (1 - \alpha) \tau_{\text{cov}} + \alpha \tau_{\text{sem}} $, where $ \alpha \in [0, 1] $ controls the emphasis between handling covariate and semantic shifts. The updated model parameters are $ \theta = \theta_{\text{pre}} + \tau $.

**Conclusion:**

By integrating LoRA modules through simple arithmetic operations, the proposed framework efficiently combines failure-specific knowledge, offering flexibility and control over the model's behavior in open environments. This method enhances the reliability of deep neural networks by simultaneously addressing covariate and semantic shifts. Addressing potential dependencies on auxiliary data and exploring scalability can further strengthen the framework, making it highly applicable in practical scenarios where robustness to various distributional shifts is essential. Moreover, the adaptable nature of this approach opens up exciting possibilities for future expansions.

**Strengths:**

- **Practical Relevance:** The method effectively addresses the challenge of unknown rejection in environments where covariate and semantic shifts occur, which is common in real-world applications.

- **Efficient Knowledge Integration:** Using LoRA modules and arithmetic operations allows for efficient integration of failure-specific knowledge without retraining the entire model, saving computational resources.

- **Flexibility and Control:** The scaling parameter$ \alpha$ enables dynamic adjustment of the model's behavior, allowing practitioners to tailor the model according to specific operational needs or preferences.

- **Comprehensive Experimental Validation:** The authors conduct extensive experiments on datasets like CIFAR-10, CIFAR-100, and ImageNet, using architectures such as ResNet and Vision Transformers. The results demonstrate that the proposed method outperforms several baselines in enhancing unknown rejection capabilities.

- **Theoretical Support:** The theoretical analysis provides insight into how the method can achieve desirable trade-offs in multi-objective optimization, reinforcing the practical findings.

**Weaknesses:**

- **Dependence on Auxiliary Data:** Reliance on Outlier Exposure means that the method requires access to auxiliary outlier data, which may not always be readily available or vary in quality, potentially affecting performance.

**Minor comments**
L239: For semantic *shifts*

**Questions:**

1. **Evaluation of Dependence on Auxiliary Data**

   - How dependent is the proposed method on the quality and availability of auxiliary outlier data used in Outlier Exposure (OE)? Can the model maintain high performance without this auxiliary data, and what alternative strategies could mitigate this dependency?

2. **Understanding Gradient Interference in Multi-Task Learning**

    - How does gradient interference in traditional multi-task learning impact the simultaneous optimization of covariate shift adaptation and semantic shift detection tasks? Could a theoretical and empirical analysis demonstrate how the proposed LoRA-based approach mitigates gradient conflicts compared to standard MTL methods?

---

> ### Author Response · Authors · 2024-11-22
> **Reply to Reviewer 8ZXm (Part 1/2 )**
>
> We sincerely appreciate Reviewer 8ZXm for the positive recommendation as well as the valuable suggestions. We really appreciate your kind words that our work is effective, efficient and flexible for unknown rejection in real-world applications. Below we would like to give detailed responses to each of your comments.
>
> _**Q1. Dependence on Auxiliary Data, and how to mitigate this dependency.**_
>
> **(1)** We agree with you that reliance on outlier exposure is a notable limitation which is also true for other OE-based OOD detection methods.  Without auxiliary data, it could be quite difficult to separate covariate shifts and semantic shifts, whose distribution often overlaps and this challenging problem is rarely considered by previous work.  While auxiliary data is important, we find that our method consistently outperforms other auxiliary-based methods such as OpenMix, SCONE, and OE (as shown in Table 1&2), which demonstrates the superiority of TrustLoRA.
>
> **(2)**  To mitigate this dependency on auxiliary data, we could consider the setting in SCONE [A], which assumes that the unlabeled covariate and semantic shifts are available at inference time and can be used to finetune the model directly. Besides, we can also explore the paradigm of human-in-the-loop to acquire few shot outliers via system feedback or active learning.  Our proposed framework can be easily extended to those settings, and more importantly, is much more efficient than existing methods since we only tune the LoRA module.
>
> **(3)** Inspired by your valuable comment, we conduct experiments to demonstrate the robustness of our method. Specifically, the RandomImage is used as auxiliary outliers following existing work. In the following Table, we show that TrustLoRA is robust to other auxiliary outliers like TIN597 [B]. We have added those results in the revised paper (Table 6 on Page 10).
>
> | Auxiliary Data       | AURC (S1) | FPR95 (S1) | AUC (S1) | F-AUC (S1) | AURC (S2) | FPR95 (S2) | AUC (S2) | F-AUC (S2) | AURC (S3) | FPR95 (S3) | AUC (S3) | F-AUC (S3) |
> |----------------------|-----------|------------|----------|------------|-----------|------------|----------|------------|-----------|------------|----------|------------|
> | **TIN597**           | 130.45    | 45.26      | 87.41    | 83.19      | 152.50    | 49.63      | 85.81    | 80.83      | 178.82    | 52.73      | 84.30    | 78.66      |
> | **RandomImage**      | 129.64    | 46.46      | 87.29    | 83.73      | 149.14    | 50.12      | 85.77    | 81.40      | 172.35    | 53.32      | 84.41    | 79.28      |
>
>
> **Reference**
>
> _[A] Feed two birds with one scone: Exploiting wild data for both out-of-distribution generalization and detection. ICML 2023_
>
> _[B] OpenOOD v1.5: Enhanced Benchmark for Out-of-Distribution Detection. ArXiv:2306.09301_

---

> ### Author Response · Authors · 2024-11-22
> **Reply to Reviewer 8ZXm (Part 2/2 )**
>
> _**Q2. Minor comments L239: For semantic shifts.**_
>
> Thank you for your careful reading, and we corrected the typo in the revised paper.
>
> _**Q3. How does gradient interference in multi-task learning impact the simultaneous optimization of covariate shift adaptation and semantic shift detection tasks? How could the proposed TrustLoRA mitigates this?**_
>
> **(1)** Conflicts existed. When optimizing both covariate shift adaptation and semantic shift detection in a multi-task learning manner, there exist remarkable conflicts between pulling covariate-shifted samples close to class centers while pushing semantic-shifted samples away from class centers. This is the reason for the poor performance of existing methods for the joint unknown rejection task. Theoretically, the Bayes-optimal reject rule for detecting misclassified covariate shifts is based on maximum class-posterior probability ${\max}_{y \in \mathcal{Y}}\mathbb{P}(y|\textbf{x})$, while detecting semantic shifts is based on density ratio $p(\textbf{x}|\text{in}) / p(\textbf{x}|\text{out})$, as follows:
>
> _Proposition_: (1) Bayes-optimal reject rule for detecting misclassified covariate shifts.  For the risk of detecting misclassified covariate shifts $R_{cov-MisD}(h,r)$, the optimal solution $r^*$ of minimizing $R_{cov-MisD}(h,r)$ is:
> $$
> r^*(x) = \mathbb{I}(\max_{y \in \mathcal{Y}}\mathbb{P}(y|{x}) \geq 1-c),
> $$
>
> where $c \in (0,1)$ is the reject cost.  (2) Bayes-optimal reject rule for detecting semantic shifts. For the risk of detecting misclassified covariate shifts $R_{OOD}(h,r)$, the optimal solution $r^*$ of minimizing $R_{OOD}(h,r)$ is:
> $$
> r^*(x) = \mathbb{I}\left(\frac{p({x}|\text{in})}{p({x}|\text{out})} > \frac{1-\pi_{\text{in}}}{\pi_{\text{in}}}\right).
> $$ where $\pi_{\text{in}} \in (0,1)$ is the mixture ratio of data of known and unknown classes.
>
> Therefore, OOD detection methods such as OE and Energy score perform density estimation explicitly or implicitly for binary discrimination of separate samples from known classes and unknown semantic-shifted classes. However, this would compress the confidence distribution of correct and incorrect covariate-shifted samples. As a result, MTL of the two objectives would be unstable.
>
> **(2)** Mitigate conflicts.  The proposed LoRA arithmetic overcomes the above limitation with reliability knowledge separation and consolidation. Specifically, we acquire the different kinds of reliability knowledge independently and then use LoRA arithmetic to get the fused model. The theoretical analysis (Proposition 3.1 on page 6) shows that TrustLoRA can flexibly find a model with a controllable solution for any preference.

---

> ### Comment · Reviewer_8ZXm · 2024-11-23
>
> Thank you for the authors' responses. I also reviewed the responses to other reviews, and I do not find any concerns significant enough to affect the score from my side, so I will keep it as is. I believe that auxiliary data, when combined with current generative AI methods, holds strong practical potential. I appreciate the good application of an existing idea like LoRA during investigating an important problem and would like to give it high contribution for that.

---

> > ### Author Response · Authors · 2024-11-23
> >
> > Dear Reviewer 8ZXm
> >
> > Thanks for your feedback. We are pleased that our responses have addressed your concerns. Thank you very much for your insightful suggestions and valuable efforts, which are crucial for enhancing the quality of our paper.
> >
> > Thank you once again.

---

### Official Review · Reviewer_Sug3 · 2024-11-05

**Soundness:** 2
**Presentation:** 2
**Contribution:** 2
**Rating:** 5
**Confidence:** 3

**Summary:**

The paper addresses the challenge of unknown rejection under both covariate and semantic shifts in deep neural models. The authors argue that existing methods for out-of-distribution (OOD) generalization and detection are misaligned with real-world applications because they either lack a rejection option or struggle with covariate shifts. TrustLoRA unifies classification with rejection by separating and consolidating failure-specific reliability knowledge using low-rank adapters, allowing for flexible and controllable unknown rejection. The framework enhances the model's ability to accept correctly classified covariate-shifted examples while rejecting misclassified ones and unknown semantic-shifted samples. Extensive experiments demonstrate TrustLoRA's superiority in handling unknown rejection compared to existing methods, showcasing its potential for safe and reliable AI applications.

**Strengths:**

- The paper proposes TrustLoRA, a simple unknown rejection framework that unifies classification with rejection under both covariate and semantic shifts, addressing a gap in real-world application requirements for deep neural models.

- TrustLoRA offers a flexible and controllable approach to unknown rejection, allowing for the adjustment of the model's reliability strength based on user preferences without the need for full retraining, which is an advantage in dynamic environments.

- The use of low-rank adapters for separating and compressing reliability knowledge is innovative, leading to a parameter-efficient method that is computationally lightweight and efficient, which is beneficial for models that need to adapt quickly to new failure cases.

- The paper provides extensive experimental results demonstrating TrustLoRA's strong performance in various scenarios, including different severities of covariate shifts and semantic shifts, showcasing its robustness and effectiveness.

**Weaknesses:**

- Although Proposition 3.1 is presented in the main paper, its proof is not provided.

- When the pre-trained model is fixed and only the LoRA module is fine-tuned, how can we ensure that the model avoids underfitting? Specifically, how can knowledge consolidation be effectively integrated into the model?

- The authors referenced [R1] in the Introduction, which also addresses covariate and semantic shifts, yet a comparative analysis with these results is absent from the experiments.

- Additional experiments on large-scale datasets, such as ImageNet, are necessary to further demonstrate the effectiveness of the proposed methods.

[R1] Feed two birds with one scone: Exploiting wild data for both out-of-distribution generalization and detection.

**Questions:**

See **Weaknesses**

---

> ### Author Response · Authors · 2024-11-22
> **Reply to Reviewer Sug3 (Part 1/2 )**
>
> We sincerely appreciate Reviewer Sug3 for the review and are grateful for the time you spent with our submission. We are glad for the acknowledgement that our approach is innovative and addresses a gap for real-world application requirements in dynamic environments. We wish to address your concerns by giving detailed responses to each of your comments as follows:
>
> _**Q1. Although Proposition 3.1 is presented in the main paper, its proof is not provided.**_
>
> As described in our theoretical analysis, Proposition 3.1 is a corollary of Theorem 4 in (Dimitriadis et al., 2023). Following your suggestion, we provide the proof below and add it to the Appendix (Section B) of the revised version.
>
> _Proof_.  Denote $\sigma$ the ReLU non-linearity $\sigma(x) = \max(0, x)$. From the universal approximation theorem [A], for any $\epsilon > 0$, there exists $Q \in \mathbb{N}$, $M \in \mathbb{R}^{(D+2) \times Q}$, $C \in \mathbb{R}^Q$, $M' \in \mathbb{R}^{Q \times D'}$ such that：
> $$\forall x \in  \mathcal{X}, \forall n \in \{1,...,N\}, ~~\|f_n(x) - g(x, \alpha) \| \leq \epsilon,$$ where $g(x, \alpha) = M'\sigma(M(x, \alpha) + C)$.
> Define two matrices $R \in \mathbb{R}^{D \times (2D+2)}$ and $S \in \mathbb{R}^{(2D+2) \times D}$ as follows:
> $R_{i,j} = \begin{cases}
> 1, & \text{if } j = 2i - 1 \\
> -1, & \text{if } j = 2i \\
> 0, & \text{otherwise}
> \end{cases}$ and   $~~~S_{i,j} = \begin{cases}
> 1, & \text{if } i = 2j - 1 \text{ or } (i > 2u \text{ and } j = u) \\
> -1, & \text{if } i = 2j \\
> 0, & \text{otherwise}
> \end{cases}$ and let $W_k = (0,...,0,k), k=1,2$.
> Then, with ${x} \in \mathbb{R}^D$, we have $$ \forall \alpha \geq 0, \quad S^\top\sigma(R^\top x + (1 - \alpha) W_1 + \alpha W_2) = ({x}, \alpha).$$
> We can learn a ReLU multi-layer perceptron $f(x, M,C,M',R,S,W)=M'\sigma(MS^\top\sigma(R^\top {x} + W) + C)$.
> Then with $\theta_{\text{pre}} = (M, C, M', R, S, 0)$, $\theta_i = (0,0,0,0,0, W_i), i=1,2$,  we have
> $$\begin{aligned}
> f({x};\theta_{\text{pre}} + (1-\alpha) \cdot \theta_1 + \alpha \cdot \theta_2) &= f({x}; M, C, M', R, S, (1 - \alpha) W_1 + \alpha W_2) \\
> &= M'\sigma(MS^\top\sigma(R^\top {x} + (1 - \alpha) W_1 + \alpha W_2 ) + C) \\
> &= g(S^\top(R^\top {x} + (1 - \alpha) W_1 + \alpha W_2)) \\
> &= g({x}, \alpha). \end {aligned}
> $$ Note that $\theta_i, i=1,2$ can be reshaped into a matrix $B_i A_i$ and in this paper we define them as  $\theta_1 = \tau_{{\rm LoRA, cov}}$ and $\theta_2 = \tau_{{\rm LoRA, sem}}$,  respectively.
>
> _**Q2. How can we ensure that the model avoids underfitting when only fine-tuning the LoRA module?**_
>
> **(1)** LoRA is a popular and effective parameter-efficient fine-tuning technique, and a higher rank $r$ means a greater number of trainable parameters and might lead to overfitting, while a lower rank $r$ means fewer trainable parameters and might lead to underfitting.   When fine-tuning a pre-trained model, if the dataset is significantly different and more complex, then it’s would be better to use a high-rank value (e.g., 64–256). On the other hand, if there doesn’t involve a complex new dataset that the model hasn’t encountered before, lower values of rank (e.g., 4–12) are sufficient.
>
> **(2)** Inspired by your valuable comment, we conduct experiments under different settings of rank in LoRA (CIFAR-10/100, Severity-1). We can find that our method is robust to the different ranks of LoRA, and we simply set $r=4$ (which is a common choice for LoRA tuning) for all experiments in our main paper. In our case, we aim to teach the model learn reliability knowledge about the current task, without introducing a complex new dataset. Therefore, a low value of rank like 4 or 2 is sufficient to learn the additional reliability knowledge without underfitting. We have added the new experiments in the appendix of the revised version.
>
> | (CIFAR-10)| AURC | FPR95 | AUC | F-AUC |(CIFAR-100)| AURC | FPR95 | AUC | F-AUC |
> |------|---------|--------|--------|--------|------|---------|--------|--------|--------|
> | $r=2$ | 29.31 | 24.92 | 93.32 | 92.24 |$r=2$ | 131.06 | 47.17 | 87.11 | 83.38 |
> | $r=4$ | 28.68 | 23.80 | 93.67 | 92.53 |$r=4$ | 129.64 | 46.46 | 87.29 | 83.73 |
> | $r=8$ | 28.45 | 23.11 | 93.60 | 92.72|$r=8$ | 128.15 | 46.03 | 87.40 | 83.82 |

---

> > ### Author Response · Authors · 2024-11-22
> > **Reply to Reviewer Sug3 (Part 2/2 )**
> >
> > _**Q3. Comparative analysis with referenced [B] is absent from the experiments.**_
> >
> > **(1)** In our initial submission, we did not compare with SCONE [B] because the setting is different. Specifically,  SCONE assumes that the real distribution of covariate and semantic shifts is available and can be used to finetune the model directly. Differently, we assume that the real distribution of covariate and semantic shifts are unavailable during the training stage, which is consistent with most of the existing studies.
> >
> > **(2)** Inspired by your comment, we adopt SCONE with outlier exposure and conduct comparable experiments with it and other more advanced methods (OpenMix, SURE, RCL, FS-KNN) suggested by Reviewer 8Zzj as follows. We can observe that our method outperforms SCONE and other methods consistently, and we have added the new experiments in Table 1 and Table 2 of the revised paper.
> >
> > | CIFAR-10        | AURC (S1) | FPR95 (S1) | AUC (S1) | F-AUC (S1) | AURC (S2) | FPR95 (S2) | AUC (S2) | F-AUC (S2) | AURC (S3) | FPR95 (S3) | AUC (S3) | F-AUC (S3) |
> > |--------------------|------------|-------------|-----------|-------------|------------|-------------|-----------|-------------|------------|-------------|-----------|-------------|
> > | OpenMix$^{\ast}$  | 29.46      | 28.13       | 92.45     | 91.08       | 46.90      | 32.61       | 90.95     | 89.12       | 66.86      | 36.94       | 89.18     | 86.83       |
> > | SURE$^{\ast}$     | 31.25      | 27.39       | 92.67     | 91.26       | 48.13      | 31.19       | 91.02     | 89.51       | 68.31      | 36.60       | 89.55     | 86.27       |
> > | RCL$^{+}$         | 58.01      | 35.92       | 89.53     | 87.47       | 93.19      | 43.11       | 86.95     | 84.17       | 132.03     | 48.36       | 84.57     | 81.12       |
> > | SCONE$^{+}$       | 44.01      | 26.99       | 93.13     | 92.08       | 71.28      | 32.75       | 91.17     | 89.37       | 104.06     | 38.91       | 88.83     | 86.15       |
> > | FS-KNN            | 47.54      | 55.75       | 87.29     | 86.01       | 62.36      | 55.35       | 86.79     | 85.30       | 80.36      | 57.86       | 85.52   | 83.74       |
> > | **TrustLoRA**     | **28.68**  | **23.80**   | **93.67** | **92.53**   | **41.64**  | **27.43**   | **92.67** | **91.18**   | **56.64**  | **30.62**   | **91.55** | **89.65**   |
> >
> > | CIFAR-100               | AURC (S1) | FPR95 (S1) | AUC (S1) | F-AUC (S1) | AURC (S2) | FPR95 (S2) | AUC (S2) | F-AUC (S2) | AURC (S3) | FPR95 (S3) | AUC (S3) | F-AUC (S3) |
> > |-----------------------|----------|-----------|---------|-----------|----------|-----------|---------|-----------|----------|-----------|---------|-----------|
> > | OpenMix$^{\ast}$ | 134.18   | 46.46     | 86.57   | 81.84     | 164.32   | 51.49     | 84.47   | 78.78     | 203.87   | 56.18     | 82.29   | 76.10     |
> > | SURE$^{\ast}$    | 137.62   | 48.35     | 86.04   | 82.26     | 172.45   | 51.76     | 83.95   | 78.41     | 205.71   | 56.62     | 82.18   | 76.54     |
> > | RCL$^{+}$        | 155.85   | 52.89     | 84.29   | 78.73     | 202.39   | 59.20     | 81.61   | 74.88     | 256.47   | 64.10     | 79.11   | 71.83     |
> > | SCONE$^{+}$     | 148.50   | 47.65     | 86.50   | 81.69     | 201.73   | 53.92     | 83.57   | 77.37     | 264.59   | 59.29     | 80.74   | 74.03     |
> > | **TrustLoRA**        | **129.64** | **46.46** | **87.29** | **83.73** | **149.14** | **50.12** | **85.77** | **81.40** | **172.35** | **53.32** | **84.41** | **79.28** |
> >
> >
> > _**Q4. Additional experiments on large-scale datasets, such as ImageNet, are necessary.**_
> >
> > To demonstrate the effectiveness of the proposed methods on large-scale datasets,  we have conducted experiments on ImageNet-200/500 with ResNet-50 in Table 4 on page 9, and the results suggest that our method yields strong unknown rejection performance compared with competitive baselines.  This experiment follows  the common practice in existing OE-like approaches which use a subset of classes in ImageNet as in distribution and others as auxiliary outliers (has no overlap with the OOD dataset at inference).
> >
> > **Reference**
> >
> > _[A] Haykin, Simon. Neural networks: a comprehensive foundation. Prentice Hall PTR, 1998._
> >
> > _[B] Feed two birds with one scone: Exploiting wild data for both out-of-distribution generalization and detection. ICML 2023._

---

> > > ### Author Response · Authors · 2024-11-25
> > > **Further comments and discussions will be appreciated**
> > >
> > > Dear Reviewer Sug3,
> > >
> > > Thank you for your valuable time to review our work and for your constructive feedback. We posted our response to your comments a few days ago with our latest experimental results, and we wonder if you could kindly share some of your thoughts so we can address your concern if there are any.
> > >
> > > In the previous response,
> > >
> > > 1. As suggested, we provided detailed proof of Proposition 3.1 and added it to the appendix.
> > >
> > > 2. We explained how to ensure that the model avoids underfitting when only fine-tuning the LoRA, and we also conducted experiments under different settings of rank in LoRA inspired by your valuable comment.
> > >
> > > 3. We added our preliminary experimental comparison with SCONE (referenced [B]) and added other more advanced methods (OpenMix, SURE, RCL, FS-KNN)  in the main Table 1&2 in the revised version.
> > >
> > > We would appreciate it if you could kindly take a look at both the revision and our response to your comments. If you have any further questions, we are happy to discuss them!
> > >
> > > Best regards,
> > >
> > > Authors

---

### Meta-Review · Area_Chair_qoML · 2024-12-17

**Metareview:**

This paper leverages LoRA to handle both covariate shifts (OOD generalization) and semantic shifts (OOD detection). This paper has divergent reviews. One of the three reviewers strongly recommended acceptance and identified exploring LoRA for OOD detection, a simple yet effective method, and the importance of addressing OOD issues as the most compelling features of the proposed method. However, other reviewers raised many concerns about novelty, missing baselines, and rigor of the theoretical analysis. At the end of the rebuttal, three of the four reviewers pushed for rejection, with several major concerns remaining, putting the paper below the acceptance bar.

**Additional Comments On Reviewer Discussion:**

This paper has received divergent reviews. One of the three reviewers strongly advocated for acceptance, while another strongly recommended rejection. The main reasons for acceptance are not compelling enough to overrule the other three reviewers who pushed for rejection, putting the paper below the acceptance bar.

---

### Decision · Program_Chairs · 2025-01-22

Reject